# Improving Robust Generalization with Diverging Spanned Latent Space

**Zhihao Dou**[*]                                        *zhihao.dou@dukekunshan.edu.cn*
*Data Science Research Center*
*Duke Kunshan University*

**Zhiqiang Gao**[*]                                        *zgao@wku.edu.cn*
*Department of Computer Science, College of Science, Mathematics and Technology*
*Wenzhou-Kean University*

**Hangchi Shen**                                        *stephen1998@email.swu.edu.cn*
*College of Artificial Intelligence*
*Southwest University*

**Ziling Yuan**                                        *leoyzl333@163.com*
*Bytedance*

**Shufei Zhang**                                        *zhangshufei@pjlab.org.cn*
*Shanghai Artificial Intelligence Laboratory*

**Kaizhu Huang**[†]                                        *kaizhu.huang@dukekunshan.edu.cn*
*Data Science Research Center*
*Duke Kunshan University*

**Reviewed on OpenReview:** *https://openreview.net/forum?id=GdaP6GgvfN*

## Abstract

Robust generalization (RG), concerning how deep neural networks could perform *over adversarial examples generated from unseen dataset*, has emerged as an active research topic. Albeit its crucial importance, most previous studies lack a well-founded theoretical analysis and certified error bounds. In this paper, we make a novel attempt to theoretically and empirically study how we could attain a better RG by learning discriminative representation, where the inconsistency of the inter-sample similarity matrix between clean and adversarial examples should be reduced. Our theoretical investigation discloses that introducing this inconsistency as a regularization term, named Gram matrix difference (GMD), will lead to tighter upper error bound and certify a better RG. Meanwhile, we demonstrate that previous efforts to reduce inter-class similarity and increase intra-class similarity among adversarial examples for enhanced adversarial robustness are approximate optimizations of our GMD approach. Furthermore, to avoid the vast optimization complexity introduced by the similarity matrix, we propose to optimize GMD by building a diverging spanned latent space for adversarial examples. On the algorithmic side, this regularization term is implemented as a novel adversarial training (AT) method — *Subspace Diverging* (SD) — to expand the volume difference between the whole latent space's linear span and subspaces' linear spans. Extensive experiments show that the proposed method can improve advanced AT methods and work remarkably well in various datasets, including CIFAR-10, CIFAR-100, SVHN, and Tiny-ImageNet.

---

[*]Equal contribution.
[†]Corresponding authors.

# 1 Introduction

Deep Neural Networks (DNNs) have attained great success recently in various applications, such as image classification, image generation, and object detection. Despite the impressive performance enhancement over various tasks, DNNs are strikingly vulnerable to specific well-crafted adversarial perturbations (Carlini & Wagner, 2017; Song et al., 2018; Fischer et al., 2017; Lyu et al., 2015). Although these perturbations are invisible to humans, they can easily mislead DNNs' predictions. Adversarial training (AT) (Mkadry et al., 2017; Wang & Zhang, 2019; Kannan et al., 2018; Gu & Rigazio, 2015; Zhang & Wang, 2019; Jia et al., 2022) is considered as one of the most effective defense methods capable of effectively improving model robustness against various types of adversarial attacks (Carlini & Wagner, 2017; Croce & Hein, 2020; Kurakin et al., 2016; Mkadry et al., 2017), such as widely used projected gradient descent (PGD) based AT (Mkadry et al., 2017), adversarial weight perturbation (AWP) (Wu et al., 2020), Feature Scatter (FS) (Zhang & Wang, 2019) and TRADES (Zhang et al., 2019).

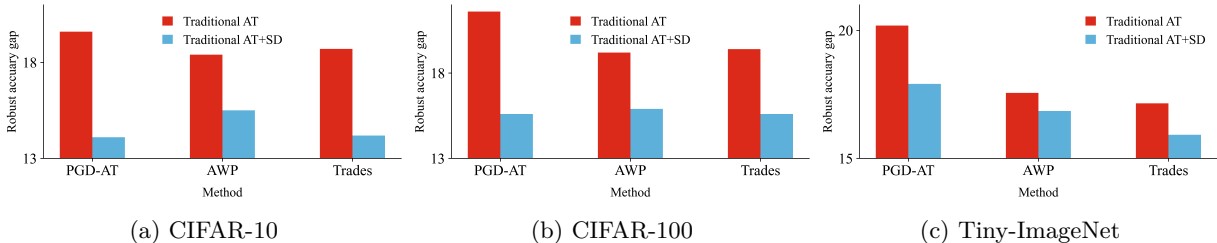

(a) CIFAR-10          (b) CIFAR-100          (c) Tiny-ImageNet

Figure 1: Robust accuracy gap (over PGD20 generated on CIFAR-10, CIFAR-100, and Tiny-ImageNet) between adversarial training and test datasets for classical AT methods and those augmented with our Subspace Diverging (SD) regularization method.

Although robustness has been improved significantly in previous studies, the robust models trained by most existing adversarial training methods still present poor robust generalization (RG). RG evaluates how well the model trained over the adversarial training set generalizes to further adversarial examples generated from unseen dataset (Rice et al., 2020), which is usually measured by the robust accuracy gap, i.e. the accuracy difference between adversarial training examples and adversarial test examples (Gao et al., 2022). As depicted in Figure 1, all robust AT models, including PGD-AT, AWP, TRADES, and FS, show poor RG and present large robust accuracy gaps.

Indeed, RG has been shown even more difficult to achieve than standard generalization since the sample complexity of adversarial examples can be significantly larger than that of standard or natural samples (Schmidt et al., 2018b). Recent efforts have studied RG from different perspectives, such as introducing customized early stopping (Rice et al., 2020), activation function (Singla et al., 2021), adversarial vertex mixup (Stutz et al., 2021), and diffusion term of Stochastic Differential Equation in AT (Sun et al., 2023). However, most of these methods empirically investigate RG and lack a well-founded theoretical analysis. Recently, Zhang et al. (2021) proposed shift consistency regularization (SCR) term to theoretically certify the RG error, which is, however, shown to be underperformed than our method in experimental results.

In this paper, from the perspective of learning robust and discriminative representation, we aim to investigate an effective algorithm that can certify a tight RG bound in theory to ensure excellent adversarial robustness practically. Generally, adversarial perturbations induce feature shifts in the latent feature representation (Zhang et al., 2021), which causes adversarial examples to move into the semantic clusters of other classes, resulting in an incorrect classification. This phenomenon can be visualized in Figure 2 by comparing the inter-sample relationship map of clean data with that of adversarial examples. In a standard model, the discriminative features of clean data are similar within the same class and dissimilar among different classes (see Figure 2a). On the contrary, in Figure 2b, features of adversarial examples exhibit smaller intra-class similarity, yet more considerable inter-class similarity than clean data. Ideally, enabling the features of adversarial examples to be as discriminative as clean data will encourage a better RG. Intuitively, this goal can be attained by reducing the inconsistency of inter-sample relationship maps between clean data and adversarial examples.

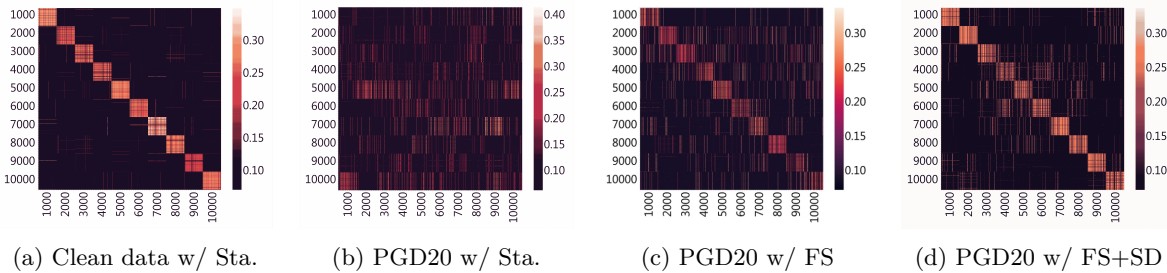

(a) Clean data w/ Sta.   (b) PGD20 w/ Sta.   (c) PGD20 w/ FS   (d) PGD20 w/ FS+SD

Figure 2: Visualization of inter-sample relationship (Gram matrix) of latent features on SVHN. We randomly select 1,000 samples for each class from the test dataset and sort all the selected samples according to their class indexes. The lighter color represents the higher cosine similarity of the two feature vectors, and vice versa. The latent features of a standard model (abbreviated to Sta.) are generated from (a) clean data and (b) adversarial examples perturbed by PGD20. The latent features of adversarial examples perturbed by PGD20 are produced by robust models trained by (c) Feature Scattering (FS) (Zhang & Wang, 2019) and (d) FS applied with our Subspace Diverging (SD). All models are trained on the SVHN dataset and employ wide residual networks (WRN-28) as the backbone.

To further verify our observation, we theoretically analyze the robust generation issue from a novel perspective. We illustrate that the robust generalization gap can indeed be upper bounded by the above inconsistency of inter-sample relationship - Gram matrix difference. Meanwhile, we demonstrate that previous efforts to reduce inter-class similarity and increase intra-class similarity among adversarial examples for enhanced adversarial robustness in (Li et al., 2019; Bui et al., 2020) are approximate optimizations of our approach. In our case, this term is optimized by learning a diverging spanned latent space for adversarial examples, where **V**olume **D**ifference between the **W**hole latent space's linear span and **S**ubspaces' linear spans (VDWS) is enlarged, such that latent subspaces (divided by categories) to be mutually orthogonal. Our method additionally improves the representation diversity of adversarial examples which has been proved as one of the merits of improving the generalization ability of standard model (Liu et al., 2018). Compared with previous studies like (Li et al., 2019; Bui et al., 2020), our method offers a theoretical guarantee for improving RG through learning discriminative representation. More importantly, compared with (Zhang et al., 2021), our approach enjoys a better RG performance, which can be observed in an empirical analysis in Section 6.6, thus certifying robust performance in many real-world datasets.

Built upon the above theory, towards better RG, we instantiate the VDWS with an AT method called *Subspace Diverging* (SD). As shown in Figure 2d, our SD improves the feature representation compared to traditional AT methods (e.g., FS in Figure 2c), which becomes more discriminative and diverse, thus promoting a better RG (as seen in Figure 1). Our contributions are digested as follows.

1) We study and reveal that the robust generalization gap correlates with the difference in inter-sample relationship maps between clean data and adversarial examples. Leveraging this insight, we derive a novel and tighter robust generalization bound.

2) To enable a tractable optimization, we propose to build a diverging spanned latent space for adversarial examples, which is theoretically well-founded for learning discriminate and diverse representation. We implement a novel adversarial regularization method named subspace diverging to achieve this goal.

3) Extensive experiments have been conducted to verify the effectiveness of our adversarial regularization method on various benchmark datasets. The results demonstrate that our approach enhances the performance of various state-of-the-art methods.

## 2 Related Work

### 2.1 Adversarial Training

Adversarial training (AT), a primary defense approach against adversarial examples (Goodfellow et al., 2015; Carlini & Wagner, 2017; Athalye et al., 2018), has been extensively researched to enhance the robustness of deep neural networks. Projected gradient descent (PGD) (Carlini & Wagner, 2017)-based AT is one of the most common approaches used to improve robustness, and PGD is an iterative optimization technique designed to generate adversarial examples by perturbing input samples within a specified norm constraint. Adversarial weight perturbation (AWP) (Wu et al., 2020) is proposed, which is effective in boosting robustness by directly perturbing the model's weights rather than the input samples. This approach aims to make the model resilient to changes in its model parameters, thereby improving its overall stability and performance against adversarial attacks. Based on AWP, Jin et al. (2023) introduces small Gaussian noise into the weights of the neural network during adversarial training, and the weight perturbation is modeled using a Taylor series expansion, which allows the method to decompose the objective function into multiple terms. The goal is to balance the trade-off between adversarial robustness and clean accuracy by smoothing the weight updates and finding flatter minima in the loss landscape. Feature Scatter (FS) (Zhang & Wang, 2019) disperses features of input data to generate diverse adversarial examples. By ensuring that adversarial examples cover a wider range of perturbations, FS can achieve better robustness. Geometry-Aware Instance Reweighted Adversarial Training (GAIRAT) (Zhang et al., 2020c) optimizes the geometry of decision boundaries by assigning different weights to adversarial examples based on their distances from the decision boundary. GAIRAT effectively balances the trade-off between robustness and accuracy, leading to significant improvements in both areas. Moreover, some researchers also investigate the effects of adversarial training strategies on model performance (Zhang et al., 2020a; Jia et al., 2022; Wei et al., 2023). Friendly Adversarial Training (FAT) (Zhang et al., 2020a) emphasizes that AT from adversarial examples closer to the decision boundary can help in reducing the model's overfitting to adversarial perturbations. Learning attack strategy (LAS) (Jia et al., 2022) is introduced to adjust the attacking configurations for different data samples, and Wei et al. (2023) adapts the class-specific training configurations.

### 2.2 Robust Generalization

Robust generalization evaluates a model's performance against unseen adversarial examples, akin to standard generalization. Yin et al. (2019.) investigate the relationship between model complexity and its ability to robust generalization against adversarial examples. It explores how increasing model capacity can improve adversarial robustness but may also lead to overfitting if not properly regularized. The finding provides theoretical insights and empirical evidence on how to balance model complexity to achieve optimal adversarial robustness and generalization. Schmidt et al. (2018b) argue that achieving robust generalization against adversarial examples necessitates significantly larger datasets compared to standard training. The statement highlights that increased data volume can enhance the model's ability to generalize and resist adversarial attacks. Zhang et al. (2019) decomposes the robust error into natural classification and boundary errors, offering a balanced approach to robustness and accuracy. This study examines the inherent trade-off between achieving robustness against adversarial attacks and maintaining high accuracy on clean data. Li et al. (2022a) analyze that the poor robust generalization is due to the VC dimension of adversarial testing samples being significantly larger than their intrinsic dimension. Sun et al. (2023) enhance robust generalization by approximating PGD-AT as a continuous-time Stochastic Differential Equation (SDE) and manipulating its diffusion term. In (Zhang et al., 2021), the Shift Consistency Regulation (SCR) method is proposed to mitigate deficient robust generalization by reducing variance in perturbation direction between adversarial training and unseen datasets. In this paper, we investigate cosine similarity variants and analyze the robust generalization gap to enhance model robustness. Theoretical analysis suggests that achieving robust generalization faces challenges from random inter-sample relationship variation, susceptible to perturbation attacks. Our approach demonstrates better performance empirically, as observed in our experiments.

### 2.3 Learning Discriminative and Diverse Representations

Metric learning-based approaches (Cheng et al., 2016b; Hadsell et al., 2006; Hu et al., 2014; Schroff et al., 2015; Huang et al., 2010; Chopra et al., 2005) are employed to increase inter-class distance and decrease intra-class distance for deep features, typically using Euclidean distance. Hadsell et al. (2006) propose how to reduce data dimensionality through invariant feature learning. It focuses on identifying and preserving essential data characteristics while minimizing irrelevant variations, thereby enhancing the efficiency and accuracy of subsequent analyses. The groundbreaking contrastive loss (Hadsell et al., 2006) enforce the above constraints using a siamese network architecture (Chopra et al., 2005). This strategy gained popularity in various downstream tasks such as image retrieval (Yousefzadeh et al., 2023). Moreover, learning discriminative feature representations can also be beneficial in face recognition (Hadsell et al., 2006; Sun et al., 2014), where the triplet loss (Cheng et al., 2016a) and center loss (Wen et al., 2016) also demonstrate similar effectiveness. In a recent work by Lezama et al. (2018), a plug-and-play loss term for deep networks has been utilized to explicitly reduce intra-class variance and enforce inter-class margin. Furthermore, Yu et al. (2020) enhanced feature representation discriminability by augmenting the code rating of feature representation. Following (Yu et al., 2020), Chan et al. (2022) introduce ReduNet, a deep learning framework that constructs interpretable network architectures by maximizing the rate reduction of feature representation. ReduNet efficiently reduces information redundancy and captures essential features.

## 3 Theoretical Analysis

This section analyzes RG from the theoretical aspect. A novel method is proposed to optimize the robust generalization gap by introducing inter-class and intra-class similarity. Moreover, we propose to leverage volume variety between the whole linear and subspace spans for a tractable optimization, which further improves the diversity of feature representation.

### 3.1 Robust Representation

Given a data distribution $(x, y) \sim (\boldsymbol{X}, \boldsymbol{Y})$ with $K$ classes, a training set, consisting of $N$ i.i.d. data pairs drawn from $(\boldsymbol{X}, \boldsymbol{Y})$, can be denoted as $(\boldsymbol{X}_D, \boldsymbol{Y}_D)$, where $\boldsymbol{X}_D \in \mathbb{R}^{d \times N}$ and $\boldsymbol{Y}_D \in \mathbb{R}^{K \times N}$ denote a training data matrix and a label matrix, respectively, and $d$ is the dimension of the data sample. The object of standard generalization is to learn a deep neural network (DNN) $f_\theta(\cdot)$ with parameters $\theta$ on a training set so that the generalization error (the difference between the expected loss over the data distribution and the empirical loss over the training data) becomes as small as possible (Xu & Mannor, 2012; Bousquet & Elisseeff, 2002; Neyshabur et al., 2017), where $f_\theta(\cdot)$ maps the data samples from the input space to the latent feature space with dimension $r$, e.g. $f_\theta(x) \in \mathbb{R}^r$. Leveraging the above insight, considering the loss function $l(\cdot)$ of $f_\theta(\cdot)$, the robust generalization error (gap) (Zhang et al., 2021) is defined as the difference between the expected loss over on adversarial examples $(\boldsymbol{X}_D^{adv}, \boldsymbol{Y}_D)$ and the expected loss over their underlying distribution $(\boldsymbol{X}^{adv}, \boldsymbol{Y})$, i.e.,

$$\varepsilon_{RGE} \triangleq \left| l\left(\boldsymbol{X}^{adv}, \boldsymbol{Y}\right) - l\left(\boldsymbol{X}_D^{adv}, \boldsymbol{Y}_D\right) \right|, \text{where}$$

$$l\left(\boldsymbol{X}^{adv}, \boldsymbol{Y}\right) = \mathbb{E}\left[ l\left(f_\theta\left(x^{adv}\right), y\right) \right],$$

$$l\left(\boldsymbol{X}_D^{adv}, \boldsymbol{Y}_D\right) = \frac{1}{N} \sum_{n=1}^{N} l\left(f_\theta(x_n^{adv}), y_n\right).$$

In general, a test set $(\boldsymbol{X}_T, \boldsymbol{Y}_T)$ is introduced as a surrogate of data distribution $(\boldsymbol{X}, \boldsymbol{Y})$ to empirically estimate the robust generalization error since the entire $(\boldsymbol{X}, \boldsymbol{Y})$ is unavailable. Here, $(\boldsymbol{X}_T, \boldsymbol{Y}_T)$ includes i.i.d. samples that are drawn from $(\boldsymbol{X}, \boldsymbol{Y})$ and disjoint with $(\boldsymbol{X}_D, \boldsymbol{Y}_D)$.

In the following, we propose Theorem 3.1 to serve as the main theoretical foundation for our work, which establishes an upper bound of the robust generalization containing standard generalization error and one novel regularization term. This term was inspired based on our observations as illustrated in Figure 2, where the Gram matrix is widely used to measure inter-class and intra-class similarity to capture inter-sample relationships. Detailed proof can be seen in the Appendix.

**Theorem 3.1** *Given the clean data matrix $\boldsymbol{X}_i$ and adversarial data matrix $\boldsymbol{X}_i^{adv}$ that both contain $N_i$ samples of $i$-th class over the training set, the sets of clean data $C_i$ and adversarial data $C_i^{adv}$ of $i$-th class over the underlying data distribution, and the DNN $f_\theta$ that maps data samples to latent features with dimension $r$, if the loss function $l(\cdot)$ of $f_\theta(\cdot)$ is $t$-Lipschitz, and $f_\theta(\cdot)$ is the $L$-Lipschitz, then for any $\sigma > 0$, with the probability at least $1 - \sigma$, we have*

$$\varepsilon_{RGE} \leq \varepsilon_{GE} + \frac{tU^2}{N}||\nabla\boldsymbol{T}_d||_2 + tKV^2||\nabla\boldsymbol{T}||_2 + tCHL||\delta||_2 + M \cdot \sqrt{\frac{2K\ln 2 + 2\ln\left(\frac{1}{\sigma}\right)}{N}},$$

$$\text{where} \quad \nabla\boldsymbol{T}_d = \boldsymbol{T}_d^{adv} - \boldsymbol{T}_d, \qquad \nabla\boldsymbol{T} = \boldsymbol{T}^{adv} - \boldsymbol{T},$$

$$U = \frac{1}{N}\sum_{n=1}^{N}||f_\theta(x_n)||_2, \quad V = \frac{1}{K}\sum_{i=1}^{K}||\mathbb{E}\left[f_\theta(x) \mid x \in C_i\right]||_2,$$

$$C = 2(N + K^3 + \frac{K^2+1}{2}), \quad H = \sup_x||f_\theta(x)||_2.$$

*$\delta$ is adversarial perturbation and $M$ is the upper bound of the loss function $l(\cdot)$ over the whole underlying data manifold. $\nabla\boldsymbol{T}_d$ and $\nabla\boldsymbol{T}$ denote the Gram matrix difference over the training set and the underlying data distribution respectively,*

$$\nabla\boldsymbol{T}_d = \begin{bmatrix} \left(\boldsymbol{Z}_1^{adv}\right)^\top \boldsymbol{Z}_1^{adv} - \left(\boldsymbol{Z}_1\right)^\top \boldsymbol{Z}_1, & \cdots, & \left(\boldsymbol{Z}_1^{adv}\right)^\top \boldsymbol{Z}_K^{adv} - \left(\boldsymbol{Z}_1\right)^\top \boldsymbol{Z}_K \\ \ddots & \ddots & \ddots \\ \left(\boldsymbol{Z}_K^{adv}\right)^\top \boldsymbol{Z}_1^{adv} - \left(\boldsymbol{Z}_K\right)^\top \boldsymbol{Z}_1, & \cdots, & \left(\boldsymbol{Z}_K^{adv}\right)^\top \boldsymbol{Z}_K^{adv} - \left(\boldsymbol{Z}_K\right)^\top \boldsymbol{Z}_K \end{bmatrix},$$

$$\nabla\boldsymbol{T} = \begin{bmatrix} \left(z_1^{adv}\right)^\top z_1^{adv} - \left(z_1\right)^\top z_1, & \cdots, & \left(z_1^{adv}\right)^\top z_K^{adv} - \left(z_1\right)^\top z_K \\ \ddots & \ddots & \ddots \\ \left(z_K^{adv}\right)^\top z_1^{adv} - \left(z_K\right)^\top z_1, & \cdots, & \left(z_K^{adv}\right)^\top z_K^{adv} - \left(z_K\right)^\top z_K \end{bmatrix}, \tag{1}$$

$$\text{where} \quad \boldsymbol{Z}_i = \frac{f_\theta\left(\boldsymbol{X}_i\right)}{||f_\theta\left(\boldsymbol{X}_i\right)||_{2,col}}, \quad \boldsymbol{Z}_i^{adv} = \frac{f_\theta\left(\boldsymbol{X}_i^{adv}\right)}{||f_\theta\left(\boldsymbol{X}_i^{adv}\right)||_{2,col}}, \quad \boldsymbol{Z}_i, \boldsymbol{Z}_i^{adv} \in \mathbb{R}^{r\times N_i},$$

$$z_i = \frac{\mathbb{E}\left[f_\theta(x) \mid x \in C_i\right]}{||\mathbb{E}\left[f_\theta(x) \mid x \in C_i\right]||_2}, \quad z_i^{adv} = \frac{\mathbb{E}\left[f_\theta\left(x^{adv}\right) \mid x^{adv} \in C_i^{adv}\right]}{||\mathbb{E}\left[f_\theta\left(x^{adv}\right) \mid x \in C_i^{adv}\right]||_2}, \quad z_i, z_i^{adv} \in \mathbb{R}^r.$$

*$||\cdot||_{2,col}$ represents the calculation of the Euclidean norm of column vectors in the matrix.*

Theorem 3.1 highlights that robust generalization gap $\varepsilon_{REG}$ can be decomposed into three terms: 1) the standard generalization gap ($\varepsilon_{GE}$); 2) *Gram matrix difference* (GMD), i.e. $\frac{tU^2}{N}||\nabla\boldsymbol{T}_d||_2 + tKV^2||\nabla\boldsymbol{T}||_2$, which measures the inter-sample relationship difference between underlying and training adversarial data distribution; 3) constant components. In GMD, $\nabla\boldsymbol{T}$ and $\nabla\boldsymbol{T}_d$ denote the Gram matrix differences on underlying and training data distributions respectively, and both of them aim to quantify the inconsistency of inter-sample relationship maps between clean and adversarial examples. To empirically illustrate the large GMD values of the vanilla AT method, we visualize the Gram matrices on a test set that is randomly sampled and includes $1,000$ data points. Figures 2a and 2c show a $\boldsymbol{T} = \left[\left(\boldsymbol{Z}_i\right)^\top \boldsymbol{Z}_j\right]$ of a standard model and a $\boldsymbol{T}^{adv} = \left[\left(\boldsymbol{Z}_i^{adv}\right)^\top \boldsymbol{Z}_j^{adv}\right]$ of a robust model trained by FS, where $i, j \in \{1, 2, \ldots, K\}$. As observed, the features of adversarial examples present smaller intra-class similarity and larger inter-class similarity than those of clean data, i.e., $||(\boldsymbol{Z}_i^{adv})^\top \boldsymbol{Z}_i^{adv}||_2 < ||\boldsymbol{Z}_i^\top \boldsymbol{Z}_i||_2$ and $||(\boldsymbol{Z}_i^{adv})^\top \boldsymbol{Z}_j^{adv}||_2 > ||\boldsymbol{Z}_i^T \boldsymbol{Z}_j||_2$. This phenomenon results in a value of Gram matrix difference, $\nabla\boldsymbol{T} = \boldsymbol{T}^{adv} - \boldsymbol{T}$, so that a better RG is hard to achieve. Accordingly, we propose to minimize the GMD to reduce this inconsistency and certify a lower RG error.

**Comparison to Shift Consistency Regularization (SCR) (Zhang et al., 2021).** According to Zhang et al. (2021), for the theoretical analysis of RG, the robust error gap is divided into three components: the standard error gap ($\varepsilon_{GE}$), a constant term, and a feature shift inconsistency term. Therefore, Zhang et al. (2021) propose SCR, which constrains the feature shift to certify RG, and SCR =

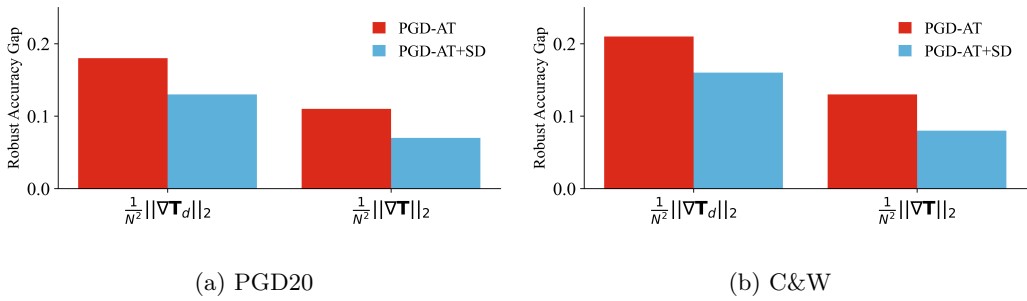

(a) PGD20          (b) C&W

Figure 3: $||\bigtriangledown \boldsymbol{T}_d||_2$ and $||\bigtriangledown \boldsymbol{T}||_2$ of our approach under (a) PGD20 and (b) C&W. All methods are trained by using WiderResNet-34-10 as backbone.

$\min_\theta \frac{t}{N} \sum_{i=1}^K \sum_{v \in \hat{N}_i} \|f_\theta(x_v^{adv}) - f_\theta(x_v) - \mathbb{E}\left[f_\theta(x^{adv}) - f_\theta(x) \mid x \in C_i\right]\|_2^2$, where $\hat{N}_i$ is the set of index of training data for class $i$. SCR only constrains the difference between the feature shift $f_\theta(x_v^{adv}) - f_\theta(x_v)$ of each training data point and the expectation of feature shifts within the same class for the underlying distribution. Nevertheless, it overlooks the inconsistency among different classes, which intuitively leads to a less tight RG compared to optimizing our GMD. We provide an indirect comparison in Section A.1 of the Appendix. We use an approximated approach to optimizing GMD, which is proposed in Section 5, and our method can achieve better RG as empirically shown in Section 6.6.

**Comparison to Inter-feature Relationship (IFR) (Zhang et al., 2024).** The approach proposed by (Zhang et al., 2024) focuses on preserving the inter-feature relationship between natural and adversarial examples. This method aims to maintain the original feature distribution structure, minimizing the variation caused by adversarial perturbations. In contrast, our approach introduces the GMD to address the inconsistency between clean and adversarial examples in the inter-sample relationships. By optimizing a diverging spanned latent space, we enhance the discriminative power of feature representations.

In terms of the optimization of GMD, we can opt to minimize $||\nabla \boldsymbol{T}_d||_2$; however, estimating $||\nabla \boldsymbol{T}||_2$ is generally impractical due to the inaccessibility of the entire underlying distribution. Alternatively, a widely accepted assumption is that the inter-feature relationships within the training data can reflect the structural information of the whole data distribution. This assumption has been effectively utilized in adversarial robustness (Li et al., 2019) and representation learning (Bui et al., 2020). Given the assumption, as the training feature representations become more discriminative, exhibiting larger intra-class similarity and smaller inter-class similarity, the feature representations of the entire underlying distribution are supposed to follow the same trend. Moreover, by learning the discriminative features of adversarial examples, the diagonal elements of the Gram matrix increase, while the off-diagonal elements decrease accordingly for both training and underlying datasets. As such, ignoring $||\nabla \boldsymbol{T}||_2$ still represents a reasonable optimization strategy, since minimizing $||\nabla \boldsymbol{T}_d||_2$ tends to cause a decrease in $||\nabla \boldsymbol{T}||_2$.

To address this aspect, we present $||\nabla \boldsymbol{T}_d||_2$ and $||\nabla \boldsymbol{T}||_2$ of our SD, as shown in Figure 3. We empirically calculate these values by randomly sampling $N = 1,000$ data points from the training and test set of the CIFAR-10 dataset respectively, where our method is only applied to the training data. $\frac{1}{N^2}||\nabla \boldsymbol{T}_d||_2$ and $\frac{1}{N^2}||\nabla \boldsymbol{T}||_2$ are calculated under the PGD20 and C&W attack with PGD-AT baseline in Figures 3a and 3b respectively. The results clearly illustrate that SD can decrease $||\nabla \boldsymbol{T}_d||_2$ and $||\nabla \boldsymbol{T}||_2$ consistently.

Moreover, GMD can be approximately optimized by minimizing $||\bigtriangledown \boldsymbol{T}_d||_2$, which can be further decomposed as:

$$\|\nabla \boldsymbol{T}_d\|_2 = \sqrt{\|\nabla \boldsymbol{T}_d^{inter}\|_2^2 + \|\nabla \boldsymbol{T}_d^{intra}\|_2^2},$$

$$\text{where} \quad \|\nabla \boldsymbol{T}_d^{inter}\|_2 = \sqrt{\sum_{i=1}^K \sum_{j=1, i \neq j}^K \|(\boldsymbol{Z}_i^{adv})^T \boldsymbol{Z}_j^{adv} - (\boldsymbol{Z}_i)^T \boldsymbol{Z}_j\|_2^2}, \tag{2}$$

$$\|\nabla \boldsymbol{T}_d^{intra}\|_2 = \sqrt{\sum_{i=1}^{K} \|(\boldsymbol{Z}_i^{adv})^T \boldsymbol{Z}_i^{adv} - (\boldsymbol{Z}_i)^T \boldsymbol{Z}_i\|_2^2};.}$$

Here, $\|\bigtriangledown \boldsymbol{T}_d^{inter}\|_2$ and $\|\bigtriangledown \boldsymbol{T}_d^{intra}\|_2$ denote the inter-class and intra-class similarity respectively on training adversarial example distribution. Reducing $\|\bigtriangledown \boldsymbol{T}_d\|_2$ can be achieved when $(\boldsymbol{Z}_i^{adv})^\top \boldsymbol{Z}_i^{adv} \longrightarrow \boldsymbol{Z}_i^\top \boldsymbol{Z}_i$ and $(\boldsymbol{Z}_i^{adv})^\top \boldsymbol{Z}_j^{adv} \longrightarrow \boldsymbol{Z}_i^\top \boldsymbol{Z}_j$. Thus, the intra-class cosine similarity should increase, and inter-class cosine similarity should decrease. This solution aligns the general intuition for learning discriminative representation to improve generalization ability and is an approximation of our GMD.

## 3.2 Diverging Spanned Latent Space

The intricacies of point-to-point level optimization contribute to the vast optimization complexity. To solve this problem, we propose to optimize this term by learning a diverging spanned latent space for adversarial examples.

The linear spans of the whole training feature matrix $\boldsymbol{Z}$ and class-wise feature matrix $\boldsymbol{Z}_i$ are denoted as $\boldsymbol{S}$ and $\boldsymbol{S_i}$ respectively, and calculated by

$$\boldsymbol{S} = \boldsymbol{Z}\boldsymbol{Z}^\top, \boldsymbol{S_i} = \boldsymbol{Z}_i\boldsymbol{Z}_i^\top,$$

where $\boldsymbol{S}, \boldsymbol{S}_i \in R^{r \times r}$, $r$ is feature dimension, and column vectors in $\boldsymbol{S}$ and $\boldsymbol{S}_i$ represent the set of basis vectors for the whole latent spaces and class-wise subspace of $i$-th class. The volume of any latent space linear span $\boldsymbol{P}$ can be represented by its determinant:

$$\text{Vol}\,(\boldsymbol{P}) = \det\,(\boldsymbol{P}). \tag{3}$$

Intuitively, the more diverse the subspaces are, the more separable clusters of adversarial example feature representation. Consequently, the inter-class similarity tends to decrease. Such a latent space, composed of diverging subspaces, embodies a larger volume, which is visualized in Figure 4. Besides, a smaller volume of each subspace indicates a larger intra-class similarity. We show the proof details of this viewpoint in Theorem 3.2.

**Theorem 3.2** Let $\boldsymbol{S}$ be the span of the latent space, encompassing all subspaces $\{\boldsymbol{S}_i\}_{i=1}^K \subset \boldsymbol{S}$. Let $\boldsymbol{I}$ denote the identity matrix. Then,

$$\begin{aligned} &\text{If} \quad \text{Vol}(\boldsymbol{S}) \quad \text{is maximized, all subspaces } \boldsymbol{S}_i \subset \boldsymbol{S} \text{ are mutually independent,} \\ &\text{If} \quad \text{Vol}(\boldsymbol{I} + \boldsymbol{S}_i) \quad \text{is minimized, the vectors in } \boldsymbol{S}_i \text{ remain consistent.} \end{aligned} \tag{4}$$

Theorem 3.2 highlights that decreased cosine similarity among distinct subspaces basis expands the volume of the overall spanned latent space as illustrated in Figure 4. In contrast, heightened cosine similarity between intra-class basis is associated with reduced volume in each subspace. Detailed proof of Theorem 3.2 is provided in the Appendix.

To achieve a discriminative feature representation, we aim to maximize the overall volume of the training latent space linear span while minimizing the volume of the linear subspace span, as elaborated in Section 3.1. Specifically, we sort these objectives by prioritizing the expansion of the **V**olume **D**ifference between the **W**hole latent space's linear span and **S**ubspaces' linear spans (VDWS):

$$VDWS \triangleq \text{Vol}\left(\boldsymbol{S}^{adv}\right) - \frac{1}{K} \sum_{i=1}^{K} \text{Vol}\left(\boldsymbol{I} + \boldsymbol{S}_i^{adv}\right), \tag{5}$$

where $\boldsymbol{I}$ is the identity matrix to prevent the subspace $S_i$ volume being close to 0; hence, the above issue can be reformulated as aiming to maximize VDWS to diminish concurrently both $\|\nabla \boldsymbol{T}_d^{inter}\|_2$ and $\|\nabla \boldsymbol{T}_d^{intra}\|_2$, denoted as:

$$\max VDWS \Longrightarrow \min \|\nabla \boldsymbol{T}_d\|_2, \tag{6}$$

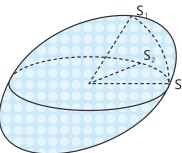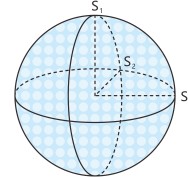

Figure 4: As the subspace becomes more independent, the volume of the overall spanned space becomes larger (each small white ball represents the unit volume, and as the white ball becomes more extensive, the overall volume becomes more prominent). Therefore, semantic clusters may become more dispersed, resulting in more discriminative feature representations.

where $||\nabla \boldsymbol{T}_d||_2$ represents the gram matrix difference on whole data distribution.

Here, we show that maximizing VDWS enjoys an additional advantage for certifying better robust generalization since the diversity of adversarial example feature representations is implicitly promoted. Specifically, VDWS that builds a diverging spanned latent space will produce a *unique eigenvector set* for each subspace, including a more significant number of eigenvectors.

By applying the singular value decomposition algorithm, the subspace can be denoted as $\boldsymbol{S}_i = \boldsymbol{U}_i \sum_i \boldsymbol{V}_i^\top$, where $\boldsymbol{U}_i$ represents the eigenvector matrix of $\boldsymbol{S}_i$. The *unique eigenvector set* for $i$-th class, including the orthogonal basis set that remains independent of other subspaces, is defined as:

$$\boldsymbol{U}_i^* = \{e_{i_1}, ..., e_{i_t}, ..., e_{i_T}\},$$
$$\text{s.t.} \quad e_{i_t} \times \boldsymbol{S}_j = 0, \quad \forall\, e_{i_t} \in \boldsymbol{U}_i^*,\ j \in [1, K] \text{ and } i \neq j. \tag{7}$$

Since our VDWS increases the volume of $\boldsymbol{S}$, the more significant number of the whole latent space basis will be attained so that the number of bases $e_{i_t}$ in unique eigenvector set $\boldsymbol{U}_i^*$ of subspace also increases.

From the above analysis, a more sizable number of distinct eigenvectors $e_{i_t}$ in $\boldsymbol{U}_{i_{adv}}^*$ will naturally steer to a greater value of $rank(\boldsymbol{U}_{i_{adv}}^*)$, which has been shown as a necessary condition for learning diverse representation in (Chan et al., 2022). Therefore, the approach not only enhances feature representation discrimination but also improves feature representation diversity, which has been proven to enhance model generalization (Liu et al., 2018). Moreover, We further verify the feature diversity of our method through experimental analysis in Section 6.8.

Previous studies have investigated the discriminative and diverse representation learning for standard generalization, such as Maximal Coding Rate Reduction (MCR$^2$), considering the low level of feature distortion as an essential premise in (Yu et al., 2020). However, as shown in Figures 2b and 2c, features of adversarial examples are highly distorted, which leads to unclear inter-sample relationships. As a result, applying MCR$^2$ directly on an adversarial example potentially introduces difficulty in the optimization process. In contrast, we propose to work on the volume to maximize VDWS to achieve robust generalization.

## 4 Empirical Analysis

In this section, we aim to confirm the validity of Theorem 3.1 by demonstrating how the accuracy gap between test and training adversarial examples evolves with varying values of VDWS. We visually illustrate inter-sample similarity, shedding light on how adversarial perturbations play a role in this accuracy gap. All the models discussed in this subsection are trained using Feature Scatter (FS) on the SVHN dataset.

## 4.1 Robust Generalization vs. Volume of Linear Span

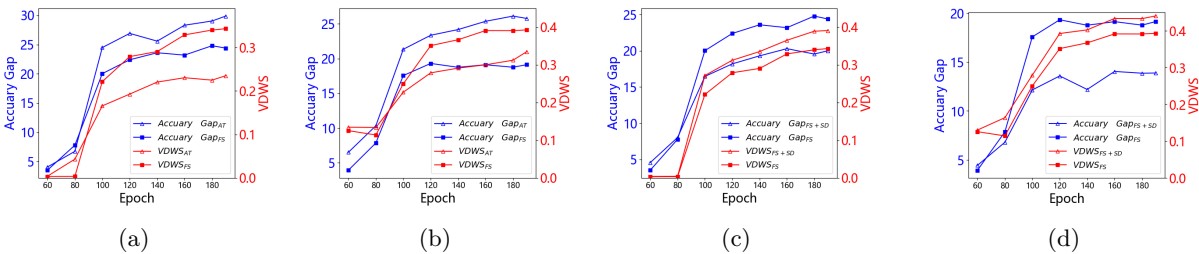

(a)  (b)  (c)  (d)

Figure 5: The accuracy gap vs. VDWS$_t$ at different training epochs. The comparison of FS and AT to train the robust models against various attacks: (a) C&W attack and (b) PGD20 attack. The comparison of FS and FS+SD (ours) to train the robust models against various attacks: (c) C&W attack and (d) PGD20 attack.

Adhering to the established evaluation protocol (Xu & Mannor, 2012), rather than using the error gap difference $|\varepsilon_{RGE} - \varepsilon_{GE}|$, we choose to calculate the accuracy gap difference, $\left| |ACC(\boldsymbol{X}_T^{adv}, \boldsymbol{Y}_T) - ACC(\boldsymbol{X}_D^{adv}, \boldsymbol{Y}_D)| - |ACC(\boldsymbol{X}_T, \boldsymbol{Y}_T) - ACC(\boldsymbol{X}_D, \boldsymbol{Y}_D)| \right|$ which shares the same trend with error gap difference. This replacement will provide a direct reflection of robust generalization, which has been widely utilized in previous studies (Zhang et al., 2021). In Section 3.1, we have demonstrated the correlation between VDWS and GMD. To validate Theorem 3.1 and show whether the gap difference is caused by GMD, the VDWS$_t$ and the accuracy gap difference under C&W and PGD20 across 60 to 200 epochs are shown in Figures 5a and 5b, where robust models are trained by FS and AT respectively and VDWS$_t$ denote the VDWS values of the test datasets. As results can be observed, the VDWS$_t$ and the accuracy gap difference show obvious consistency. Therefore, the results indicate that GMD can capture the error gap difference (reflecting the difference between robust and standard generalization).

**Feature Visualization**

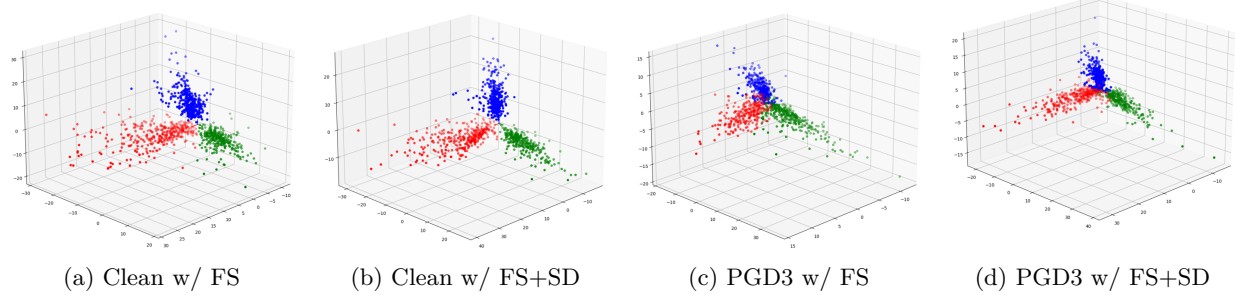

(a) Clean w/ FS  (b) Clean w/ FS+SD  (c) PGD3 w/ FS  (d) PGD3 w/ FS+SD

Figure 6: The latent features generated from (a) FS with clean samples, (b) FS+SD with clean samples, (c) FS against PGD3 attack, (d) FS+SD against PGD3 attack. All models are trained on the CIFAR-10 dataset and employ wide residual networks (WRN-28) as the backbone.

We select three classes for visualizing the feature distributions on the test dataset of CIFAR-10. We employ a multi-layer perceptron (MLP) backbone and utilize a PGD-3 attack with $\epsilon = \frac{2}{255}$ to generate feature distributions presented in Figure 6. In the latent space of adversarial examples, after integrating our SD to FS in Figure 6d, the distributions of the categories are more inclined toward the vertical, and the clusters of categories are more compact than the ones of FS in Figure 6c. This trend also emerges in the clean dataset, as observed in models trained by FS+SD in Figure 6b and FS in Figure 6a.

# 5 Subspace Diverging Regularization

We now turn to solve the optimization problem outlined in Section 3.2. Initially, to expand the entire volume of the latent linear span $\text{Vol}\,(\boldsymbol{S})$, we introduce the following functions:

$$L_{\text{span}} = \log\det(\boldsymbol{S}). \tag{8}$$

The function $\log\det(\cdot)$ is a smoothly concave function that aids in achieving an optimal solution more effectively. Moving on, to reduce the volume of the subspace linear span, we define the contraction component as below:

$$L_{\text{shrink}} = \sum_{i=1}^{K} \frac{N_i}{2N} \log\det\left(\boldsymbol{I} + \frac{\omega}{N_i}\boldsymbol{S_i}\right), \tag{9}$$

where $N_i$ represents the number of training samples for each class, and $\omega$ is a pre-defined parameter. These two components collaboratively function as:

$$L_{\text{diverge}} = \gamma L_{\text{shrink}} - (1-\gamma)L_{\text{span}}, \tag{10}$$

where $\gamma$ is a balance hyper-parameter, scaling two functions effectively.

---

**Algorithm 1** Adversarial Training with our Subspace Diverging (SD)

---

**Input:** a neural network $f_\theta(\cdot)$ initialized with learnable parameters $\theta$, $\hat{n}$ batches of data pairs $\{(\hat{x}_1, \hat{y}_1), (\hat{x}_2, \hat{y}_2), \ldots, (\hat{x}_{\hat{n}}, \hat{y}_{\hat{n}})\}$, batch size V, a predefined hyper-parameter $\lambda$, Gaussian noise $\epsilon \in (-0.015, 0.015)$, number of epochs $e$.
**Output:** robust neural network $f_\theta(\cdot)$.
Initialize $L_{gene}^{max} \leftarrow 0$, $L_{diverge}^{max} \leftarrow 0$, $L^{max} \leftarrow 0$
**for** $i \leftarrow 1$ to $\hat{n}$ **do**
    Add Gaussian noise on data samples: $\hat{x}_i' = \hat{x}_i + \epsilon$
    Generate the adversarial examples for $i$-th data batch: $\hat{x}_i^{\text{adv}} \leftarrow \underset{\hat{x}_i'}{\text{argmax}}\,(L_{\text{gene}}(\hat{x}_i', \hat{y}_i, \theta))$

    Select the maximum loss values for three sets:
        $L_{gene}^{max} \leftarrow \max(L_{gene}(\hat{x}_i', \hat{y}_i, \theta), L_{gene}^{max})$
        $L_{diverge}^{max} \leftarrow \max(L_{diverge}(\hat{x}_i', \hat{y}_i, \theta), L_{gene}^{max})$
        $L^{max} \leftarrow \max(L(\hat{x}_i^{adv}, \hat{y}_i, \theta), L^{max})$
**end for**
Calculate the $k$ and $d$ by: $k = \frac{L_{diverge}^{max}}{L^{max}}$ and $d = \frac{L_{diverge}^{max}}{L_{gene}^{max}}$
**for** $j \leftarrow 1$ to $e$ **do**
    **for** $i \leftarrow 1$ to $\hat{n}$ **do**
        $\hat{x}_i' \leftarrow \hat{x}_i + \epsilon$
        Generate the adversarial examples: $\hat{x}_i^{\text{adv}} \leftarrow \underset{\hat{x}_i'}{\arg\max}\,\left(L_{\text{gene}}(\hat{x}_i', \hat{y}_i, \theta) + \frac{1}{d}L_{\text{diverge}}(\hat{x}_i', \hat{y}_i, \theta)\right)$
        Update the classifier: $\theta \leftarrow \underset{\theta}{\min}\left\{\frac{1}{V}\left(L(\hat{x}_i^{\text{adv}}, \hat{y}_i, \theta) + \frac{\lambda}{k}L_{\text{diverge}}(\hat{x}_i^{\text{adv}}, \hat{y}_i, \theta)\right)\right\}$
    **end for**
**end for**
**return** robust neural network $f_\theta(\cdot)$.

---

Consequently, we propose utilizing $L_{\text{diverge}}$ as a regularization term in adversarial training. This regularization promotes subspace divergence, which can be defined as:

$$\min_\theta \{\frac{1}{N}L\left(\boldsymbol{X}_D^{adv}, \boldsymbol{Y}_D, \theta\right) + \frac{\lambda}{k} * L_{\text{diverge}}\left(\boldsymbol{X}_D^{adv}, \boldsymbol{Y}_D, \theta\right)\},$$

$$\text{s.t. } \boldsymbol{X}_D^{adv} = \underset{X_D'}{\text{argmax}}\left(L_{gene}\left(\boldsymbol{X}_D', \boldsymbol{Y}_D, \theta\right) + \frac{1}{d}L_{\text{diverge}}\left(\boldsymbol{X}_D', \boldsymbol{Y}_D, \theta\right)\right),$$

Table 1: All used hyper-parameters on different methods.

| Method | Datasets | Batch Size | Training Epoch | (d,k) | $\gamma$ | $\lambda$ | $\omega$ |
|---|---|---|---|---|---|---|---|
| PGD-AT+SD | CIFAR-10 | 120 | 400 | (100,100) | 0.5 | 0.5 | 0.05 |
| | CIFAR-100 | 80 | 300 | (200,200) | 0.7 | 0.05 | 0.01 |
| | Tiny-ImageNet | 120 | 120 | (200,400) | 0.2 | 0.01 | 0.05 |
| AWP+SD | CIFAR-10 | 120 | 200 | (100,100) | 0.5 | 0.5 | 0.1 |
| | CIFAR-100 | 120 | 250 | (200,200) | 0.7 | 0.02 | 0.1 |
| | Tiny-ImageNet | 120 | 130 | (150,400) | 0.3 | 0.01 | 0.5 |
| FS+SD | CIFAR-10 | 120 | 300 | (200,100) | 0.5 | 0.5 | 0.1 |
| | CIFAR-100 | 100 | 400 | (200,150) | 0.5 | 0.02 | 0.05 |
| | SVHN | 120 | 600 | (200,100) | 1 | 0.05 | 0.05 |
| TRADES+SD | CIFAR-10 | 80 | 90 | (90,50) | 0.8 | 0.1 | 0.05 |
| | CIFAR-100 | 100 | 90 | (150,200) | 0.5 | 0.01 | 0.01 |

where $\boldsymbol{X}_D^{adv}$ indicate adversarial training dataset that are produced during the adversarial training, $\boldsymbol{X}_D'$ represents the initial adversarial training dataset that adds random noise to the original image, $L(\cdot)$ is an adversarial training loss, and $L_{gene}(\cdot)$ is the loss function which producing adversarial examples. As the scale of $L_{diverge}(\cdot)$ and $L_{gene}(\cdot)$ are different, we design normalization parameters $d$ and $k$ to adjust loss values, and $\lambda$ is an additional pre-defined parameter.

Due to the GPU memory limitations, calculating $L_{diverge}(\cdot)$ for all of the training dataset is very challenging. To solve this problem, we calculate $L_{diverge}(\cdot)$, $L(\cdot)$, and $L_{gene}(\cdot)$ for each data batch. Then, we select the maximum values among all the batches and define them as $L^{max}$, $L_{gene}^{max}$, and $L_{diverge}^{max}$ respectively. Subsequently, we estimate $d$ and $k$ using $L^{max}$, $L_{diverge}^{max}$ and $L_{gene}^{max}$, where $d = \frac{L_{diverge}^{max}}{L_{gene}^{max}}$ and $k = \frac{L_{diverge}^{max}}{L^{max}}$.

We have observed that the values of $d$ and $k$ remain stable for each epoch. Therefore, to streamline our training process, we compute the $d$ and $k$ only in the first epoch. The specific values for $d$ and $k$ of various datasets and the entire algorithm can be found in experiment section. When training the generated adversarial examples, our algorithm can be regarded as a regularization to increase the overall spanned space volume but reduce the volume of each subspace.

## 6 Experiments

This section conducts comprehensive experiments to gauge the effectiveness of our SD method, in countering diverse adversarial examples.

### 6.1 Experimental Setting

We evaluate our method's robustness against white-box and black-box adversarial examples on CIFAR-10, CIFAR-100, SVHN, and Tiny-Imagenet. We benchmark our approach against four established methods: Feature Scattering (FS) (Zhang & Wang, 2019), Adversarial Training (AT) (Mkadry et al., 2017), Adversarial Weight Perturbation (AWP) (Wu et al., 2020), and TRADES (Zhang et al., 2019). Our core model is based on the WideResNet-34-10 (WRN34) architecture. On Tiny-ImageNet datasets, we follow (Jia et al., 2022) and implement PreActResNet18 as the backbone. FS, a well-known baseline, has been demonstrated to perform poorly against strong attacks, such as Autoattack, in previous work (Naseer et al., 2022). As a result, we do not present it as a primary outcome. However, since SCR is built upon FS, we will compare SD with SCR based on FS baseline alone in Section 6.3. Following (Zhang et al., 2021), we implement WideResNet-28-10 (WRN28) for comparison.

In our training regimen, we employ SGD with a momentum of 0.9, weight decay of $5 \times 10^{-4}$, and an initial learning rate of 0.1. Learning rates decrease at epochs 60 and 90 by a factor of 0.1. During training, we perform 7 attack iterations for PGD-AT and TRADES, and 1 iteration for FS. For consistency, the attack budget $\epsilon$ is maintained at 8/255 for all the methods. Adversarial examples are computed with the $\ell_\infty$ norm during training and testing. All experiments are conducted on a single GPU, e.g. RTX 3090, under the

Table 2: Accuracy under white-box attacks on CIFAR-10 with WiderResNet-34-10 ($\epsilon = 8$).

| Method | Clean | PGD-20 | PGD-50 | C&W | AA |
|---|---|---|---|---|---|
| PGD-AT | 85.17 | 55.08 | 54.88 | 53.91 | 51.69 |
| TRADES | 85.72 | 56.10 | 55.90 | 53.87 | 53.40 |
| AWP | 85.57 | 58.13 | 57.92 | 56.03 | 53.90 |
| LBGAT | 88.22 | 54.66 | 54.30 | 54.29 | 52.23 |
| MART | 84.17 | 58.56 | 58.06 | 54.58 | 51.10 |
| FAT | **87.97** | 49.86 | 48.79 | 48.65 | 47.48 |
| GAIRAT | 86.30 | 59.54 | 58.74 | 45.57 | 40.30 |
| PGD-AT+LAS | 86.23 | 56.49 | 56.12 | 55.73 | 53.58 |
| PGD-AT+SCR | 85.91 | 56.91 | 56.51 | 54.93 | 53.04 |
| PGD-AT+RAT | 84.39 | 56.29 | 55.97 | 55.18 | 52.38 |
| TRADES+LAS | 85.24 | 57.07 | 56.80 | 55.45 | 54.15 |
| TRADES+SCR | 86.31 | 56.81 | 55.97 | 54.29 | 54.10 |
| TRADES+RAT | 85.98 | 58.47 | - | 56.13 | 54.20 |
| AWP+LAS | 87.74 | 60.16 | 59.79 | 58.22 | 55.52 |
| AWP+SCR | 85.49 | 60.90 | 58.31 | 56.36 | 53.49 |
| AWP+RAT | 86.12 | 61.45 | - | 58.22 | 57.40 |
| **PGD-AT+SD** | $86.43 \pm 0.12$ | $58.93 \pm 0.23$ | $58.29 \pm 0.18$ | $55.79 \pm 0.22$ | $53.91 \pm 0.16$ |
| **TRADES+SD** | $85.99 \pm 0.14$ | $58.79 \pm 0.24$ | $57.03 \pm 0.22$ | $56.93 \pm 0.17$ | $55.03 \pm 0.15$ |
| **AWP+SD** | $87.79 \pm 0.11$ | $\mathbf{61.59} \pm 0.19$ | $\mathbf{60.30} \pm 0.21$ | $\mathbf{58.73} \pm 0.18$ | $\mathbf{57.91} \pm 0.20$ |

Table 3: Accuracy under white-box attacks on CIFAR-100 with WiderResNet-34-10 ($\epsilon = 8$).

| Method | Clean | PGD-20 | PGD-50 | C&W | AA |
|---|---|---|---|---|---|
| PGD-AT | 60.89 | 31.69 | 31.45 | 30.10 | 27.86 |
| TRADES | 58.61 | 28.66 | 28.56 | 27.05 | 25.94 |
| AWP | 60.38 | 33.86 | 33.65 | 31.12 | 28.86 |
| LBGAT | 60.64 | 34.75 | 34.62 | 30.65 | 29.33 |
| SAT | 62.82 | 27.17 | 26.76 | 27.32 | 24.57 |
| PGD-AT+LAS | 61.80 | 32.77 | 32.54 | 31.12 | 29.03 |
| PGD-AT+SCR | 60.90 | 32.97 | 31.58 | 30.39 | 29.17 |
| PGD-AT+RAT | 61.15 | 32.29 | 30.97 | 31.53 | 28.77 |
| TRADES+LAS | 60.62 | 32.53 | 32.39 | 29.51 | 28.12 |
| TRADES+SCR | 60.42 | 32.15 | 32.17 | 29.57 | 27.68 |
| TRADES+RAT | 62.93 | 33.36 | - | 29.61 | 27.90 |
| AWP+LAS | 64.89 | 36.36 | **36.13** | 33.92 | 30.77 |
| AWP+SCR | 64.51 | 34.92 | 33.98 | 33.58 | 29.79 |
| AWP+RAT | 64.71 | 35.73 | - | 31.41 | 30.20 |
| **PGD-AT+SD** | $62.34 \pm 0.21$ | $33.19 \pm 0.15$ | $32.97 \pm 0.20$ | $31.99 \pm 0.19$ | $29.58 \pm 0.18$ |
| **TRADES+SD** | $60.97 \pm 0.16$ | $33.79 \pm 0.22$ | $33.17 \pm 0.19$ | $29.79 \pm 0.17$ | $28.43 \pm 0.14$ |
| **AWP+SD** | $\mathbf{64.91} \pm 0.14$ | $\mathbf{36.59} \pm 0.17$ | $35.97 \pm 0.16$ | $\mathbf{34.03} \pm 0.19$ | $\mathbf{31.14} \pm 0.17$ |

environment using CUDA 11.7, Python 3.8, and Pytorch 1.80. Table 1 lists all used parameters on different baselines, batch sizes, and training epochs. $\gamma$ is the balance parameter, $\omega$ and $\lambda$ are pre-defined parameters, $d$ and $k$ are normalization parameters. The early stopping is used for model selection in all methods. Our results and standard deviations are obtained from 5 runs; each trained using a different random seed.

Based on various baselines, we compare our proposed baseline+SD approach with other state-of-the-art adversarial training methods : 1) PGD-AT (Mkadry et al., 2017), 2) AWP (Wu et al., 2020), 3) FS (Zhang & Wang, 2019). 4) SCR (Zhang et al., 2021), 5) LAS(Jia et al., 2022), 6) GAIRAT (Zhang et al., 2020c), 7) SAT (Sitawarin et al., 2021), 8) FAT (Zhang et al., 2020b), 9) LAT (Kumari et al., 2019), 10) Bilateral (Wang & Zhang, 2019) and 11) RAT (Jin et al., 2023). We report the results under the white-box attack in Table 2, Table 3, Table 4, and Table 5 while leaving the results of the black-box attack in Table 7.

## 6.2 Robustness Against Adversarial Examples

Our results reveal that SD can improve the robustness of different baselines against attacks on various datasets. Even for more complex datasets such as Tiny-imagine, SD still has a significant effect compared with baselines. Tables 2-5 summarize the robust accuracy of different methods under various adversarial attacks across CIFAR-10, CIFAR-100, Tiny-ImageNet, and SVHN datasets. Our method, SD, consistently achieves superior performance, improving both accuracy and robustness across all datasets and attack types. Notably, SD demonstrates significant improvements over baseline methods, particularly in challenging scenarios such as the PGD and C&W attacks. These results highlight the effectiveness of SD in enhancing model robustness, making it a reliable solution for adversarial defense.

Table 4: Accuracy under white-box attack on Tiny-Imagenet ($\epsilon = 8$).

| Method | Clean | PGD20 | PGD50 | C&W | AA |
|---|---|---|---|---|---|
| PGD-AT | 41.98 | 20.43 | 19.98 | 17.60 | 13.78 |
| AWP | 41.48 | 22.79 | 22.51 | 19.02 | 17.34 |
| PGD-AT+LAS | 44.86 | 22.29 | 22.16 | 18.54 | 16.74 |
| AWP+LAS | 45.26 | 23.77 | 23.42 | 19.88 | 18.42 |
| PGD-AT+SD | $44.27 \pm 0.13$ | $23.15 \pm 0.21$ | $22.97 \pm 0.18$ | $18.59 \pm 0.19$ | $16.79 \pm 0.17$ |
| AWP+SD | $\mathbf{45.59} \pm 0.11$ | $\mathbf{23.91} \pm 0.16$ | $\mathbf{23.49} \pm 0.19$ | $\mathbf{20.07} \pm 0.15$ | $\mathbf{18.81} \pm 0.14$ |

Table 5: Accuracy under white-box attacks on SVHN ($\epsilon = 8$).

| Models | Clean | PGD20 | PGD100 | C&W |
|---|---|---|---|---|
| Standard | **97.20** | 0.30 | 0.10 | 0.30 |
| PGD-AT | 93.90 | 47.90 | 46.00 | 48.70 |
| LAT | 91.65 | 60.23 | 59.97 | - |
| Bilateral | 94.10 | 53.90 | 50.30 | - |
| FS | 96.20 | 62.90 | 52.00 | 61.30 |
| FS+SCR | 96.60 | 70.24 | 60.72 | 64.42 |
| **FS+SD** | $97.10 \pm 0.27$ | $\mathbf{74.15} \pm 0.22$ | $\mathbf{66.57} \pm 0.19$ | $\mathbf{65.91} \pm 0.22$ |

## 6.3 Results on FS Baseline Compared with SCR

Since SCR is based on the FS baseline, we added an extra section to compare SCR and SD specifically based on the FS baseline. Table 6 presents the performance of different methods based on the FS baseline on CIFAR-10, CIFAR-100 and SVHN. Notably, the results marked with an asterisk (*) are cited from (Naseer et al., 2022). On CIFAR-10, SD performs worse than SCR against certain attacks, such as PGD20. However, on CIFAR-100 and SVHN, SD comprehensively outperforms SCR. We demonstrated improved robust accuracy with SD compared to SCR across nearly all baselines, datasets, and attack types.

Table 6: Accuracy under white-box attack on FS baseline with WiderResNet-28-10 ($\epsilon = 8$).

| Dataset | Method | Clean | PGD-20 | PGD-100 | C&W | AA |
|---|---|---|---|---|---|---|
| CIFAR-10 | FS | 90.00 | 70.50 | 68.60 | 62.40 | 36.64 |
| | FS+SCR | **92.70** | **76.45** | 67.79 | **75.42** | 35.81 |
| | FS+SD | $91.91 \pm 0.14$ | $72.33 \pm 0.18$ | $\mathbf{71.85} \pm 0.22$ | $68.66 \pm 0.19$ | $\mathbf{37.85} \pm 0.16$ |
| CIFAR-100 | FS | 73.90 | 47.20 | 46.20 | 34.60 | 0.00* |
| | FS+SCR | 74.20 | 48.87 | 47.34 | 38.90 | - |
| | FS+SD | $\mathbf{74.55} \pm 0.12$ | $\mathbf{51.03} \pm 0.17$ | $\mathbf{49.57} \pm 0.16$ | $\mathbf{41.53} \pm 0.19$ | $\mathbf{4.77} \pm 0.14$ |
| SVHN | FS | 96.20 | 62.90 | 52.00 | 61.30 | 25.26 |
| | FS+SCR | 96.60 | 70.24 | 60.72 | 64.62 | - |
| | FS+SD | $\mathbf{97.10} \pm 0.11$ | $\mathbf{74.15} \pm 0.15$ | $\mathbf{66.57} \pm 0.19$ | $\mathbf{65.91} \pm 0.20$ | $\mathbf{33.51} \pm 0.17$ |

## 6.4 Results on Different Attack Budget

Figures 7 and 8 illustrate the effectiveness of SD, our proposed method, in improving model robustness under both PGD20 and CW attacks for CIFAR-10 and CIFAR-100 datasets. Across all attack budgets, models incorporating SD show a more gradual decline in accuracy, particularly for PGD-AT+SD. The incorporation of SD enhances the resilience of both PGD-AT and TRADES, with PGD-AT+SD demonstrating the most significant improvement, especially as the attack budget increases. This suggests that SD effectively promotes robustness by maintaining higher accuracy under stronger attack conditions.

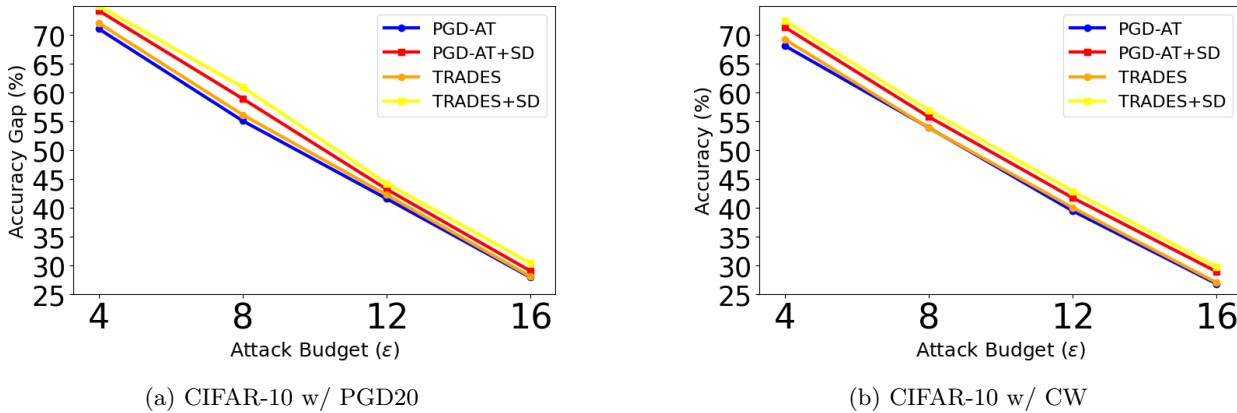

(a) CIFAR-10 w/ PGD20          (b) CIFAR-10 w/ CW

Figure 7: Accuracy vs. different attack budget for CIFAR-10 under PGD20 and CW attacks.

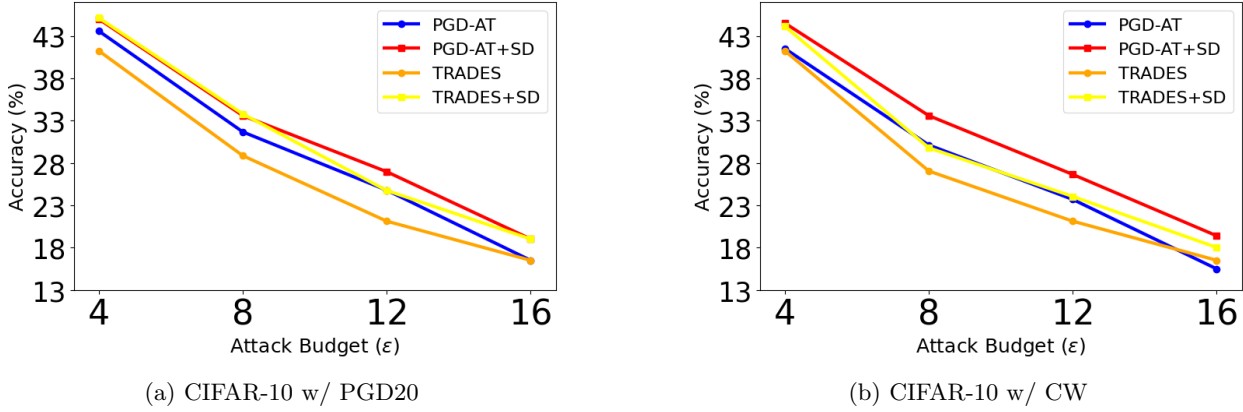

(a) CIFAR-10 w/ PGD20          (b) CIFAR-10 w/ CW

Figure 8: Accuracy vs. different attack budget for CIFAR-100 under PGD20 and CW attacks.

## 6.5 Results on Black-Box Attack

Table 7 presents the robust accuracies under black-box attacks on three datasets, including CIFAR-10, CIFAR-100, and Tiny-ImageNet. Here, since the compared methods are evaluated by using different backbones in original papers, it is difficult to compare the results of black-box attacks directly. To ensure a fair comparison, we retrain all compared methods by leveraging the WiderResNet-34-10 (WRN34) as the unified backbone and maintaining their original training parameter. We implement two black-box attacks FAB and SQUARE belonging to Autoattack. By consistently outperforming other methods, particularly under black-box attacks, SD proves to be a robust and reliable defence mechanism. The significant improvements in accuracy across CIFAR-10, CIFAR-100, and Tiny-ImageNet confirm that SD offers superior protection.

Table 7: Accuracy under transfer-based black-box attack on various datasets ($\epsilon = 8$).

| Method | CIFAR-10 | | CIFAR-100 | | Tiny-ImageNet | |
|---|---|---|---|---|---|---|
| | $\mathbf{AA_{FAB}}$ | $\mathbf{AA_{SQUARE}}$ | $\mathbf{AA_{FAB}}$ | $\mathbf{AA_{SQUARE}}$ | $\mathbf{AA_{FAB}}$ | $\mathbf{AA_{SQUARE}}$ |
| AWP | 60.51 | 61.28 | 30.97 | 35.88 | 17.19 | 22.91 |
| AWP+LAS | 62.41 | 63.39 | 31.20 | 37.29 | 17.82 | 24.42 |
| AWP+RAT | 61.75 | 63.25 | 30.09 | 36.10 | **18.16** | 22.98 |
| AWP+SCR | 62.01 | 62.29 | 31.10 | 35.49 | 17.52 | 23.61 |
| **AWP+SD** | $\mathbf{63.02} \pm 0.15$ | $\mathbf{63.88} \pm 0.18$ | $\mathbf{31.72} \pm 0.19$ | $\mathbf{37.90} \pm 0.16$ | $17.73 \pm 0.12$ | $\mathbf{24.49} \pm 0.14$ |

## 6.6 Generalization Analysis

Figures 5c and 5d illustrate the robust accuracy gap under C&W and PGD20 between training and test datasets of CIFAR-10. We present VDWS$_t$ for both the FS and FS+SD. Notably, Figures 5c and 5d highlight that FS+SD achieves a smaller accuracy gap and a greater VDWS$_t$ in comparison to FS. FS+SD consistently maintains superior VDWS$_t$ throughout the convergence process and exhibits a reduced accuracy gap. This observation underscores SD's role in diminishing the accuracy gap and enhancing robust generalization by emphasizing the volume difference between the entire space and the summation of subspaces.

Figures 9a and 9b illustrate a comparison of RG results obtained through optimization using the SD regularization term versus the SCR regularization term. We use PGD-AT as a baseline and attacks are PGD20 and C&W, respectively. It is evident that using the SD regularization term results in a smaller RG compared to the SCR regularization term. The reason is that compared to SCR, SD not only constrains the inter-sample relationship variations within the same class but also decreases the inter-class similarity. The outcomes validate that SD can achieve better RG compared with SCR term.

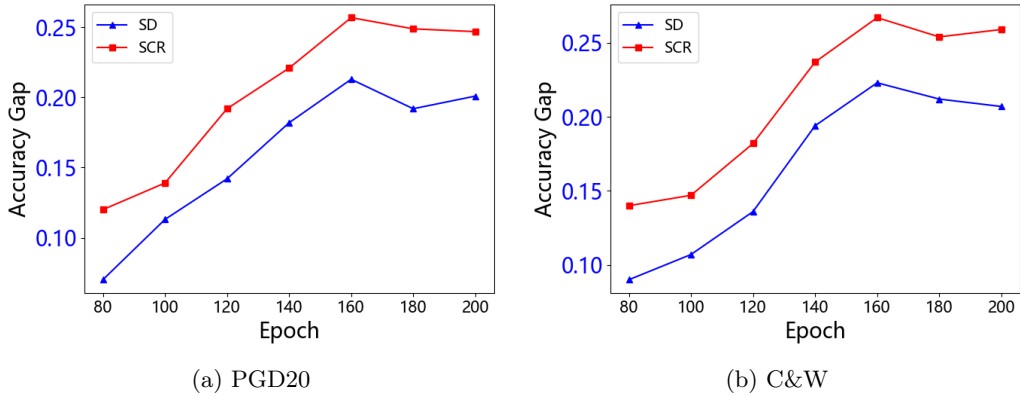

(a) PGD20  (b) C&W

Figure 9: RG with PGD-AT+SD and PGD-AT+SCR on CIFAR-100 under (a) PGD20 and (b) C&W. All methods are trained by using WiderResNet-34-10 as backbone.

## 6.7 Meta-Analysis

In this subsection, we present a meta-analysis comparing the performance of different adversarial training methods and our SD regularization term, focusing on robust accuracy under PGD20 attack across CIFAR10 and CIFAR100. To rigorously quantify the differences in performance, we employed two statistical measures: Cohen's d effect size and t-test. Cohen's d measures the effect size, which quantifies the magnitude of the improvement in robust accuracy when SD is added. t-test assess whether the differences in robust accuracy between methods with and without SD are statistically significant. This combined approach helps us evaluate both how much SD improves robustness and whether these improvements are consistent across datasets.

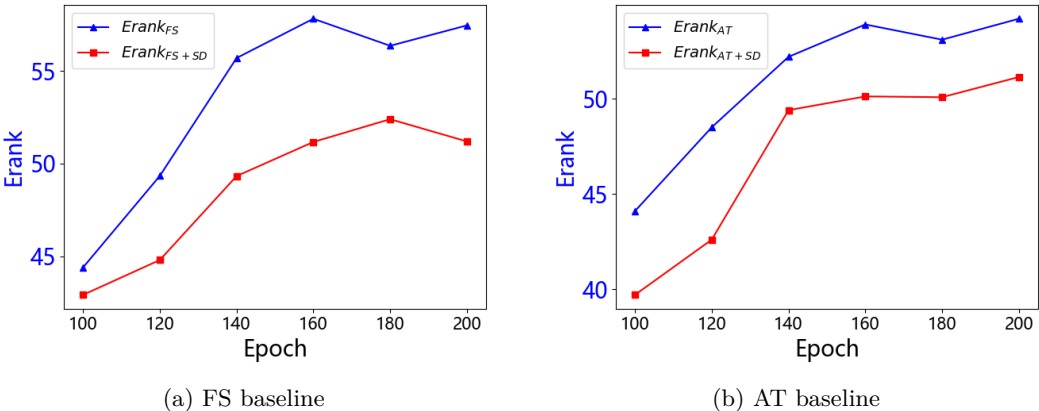

(a) FS baseline             (b) AT baseline

Figure 10: *Erank* values under PGD20 attacks from the CIFAR-100 dataset.

Table 8: Robust accuracy of different baselines on various datasets under PGD-20 attacks.

| Method | CIFAR-10 | CIFAR-100 |
|---|---|---|
| PGD-AT | 55.08 | 31.69 |
| AWP | 58.13 | 33.86 |
| TRADES | 56.10 | 28.66 |
| PGD-AT + SD | 58.93 | 33.19 |
| AWP + SD | **61.59** | **36.59** |
| TRADES + SD | 58.79 | 33.79 |

Table 9: Cohen's d for AWP vs. AWP + SD across datasets and t-test results for PGD-AT vs. PGD-AT + SD

| Dataset | Cohen's d (AWP vs. AWP + SD) | t-test (PGD-AT vs. PGD-AT + SD) |
|---|---|---|
| CIFAR-10 | 0.78 | $p < 0.05$ |
| CIFAR-100 | 0.65 | $p = 0.03$ |

The results in Table 8 show that SD regularization significantly improves robust accuracy when combined with adversarial training methods. In Table 9, Cohen's d values confirm moderate to large effects across two datasets, indicating a meaningful impact. In conclusion, SD enhances both the generalization and robustness of adversarial training, making it a valuable addition to defenses against adversarial attacks.

## 6.8 Diversity Analysis

According to the definition given in Equation 7, an increase in the total unique vector numbers reflects greater feature representation diversity. However, the computation of unique eigenvector numbers poses a challenging task. To facilitate direct measurement of the diversity within the spanned space $\boldsymbol{S}$, we utilize a widely adopted evaluation metric, effective rank, denoted as $Erank$. It can be mathematically represented as:

$$Erank(\boldsymbol{S}) = exp\left(-\sum_{l=1}^{L} p_l log\left(p_t\right)\right), \quad p_l = \frac{\alpha_l}{||\alpha||_1}, \tag{11}$$

where $\alpha_l$ denote the $l$-th element in an eigenvalue set $\alpha = \{\alpha_1, \dots, \alpha_l, \dots, \alpha_L\}$ of the spanned space $\boldsymbol{S}$, $||\alpha||_1$ represents the L1 norm of $\alpha$, and $p_l$ is the normalized eigenvalue. The larger $Erank$ represents the better diversity for feature representation. When the values of $Erank$ reach maximum, the determinant of $\boldsymbol{S}$ (i.e. $\det(\boldsymbol{S}) = \prod_{l=1}^{L} \alpha_l$) also reach maximum. Following the above analysis, we can conclude that there is a positive relationship between VDWS and $Erank$. Increasing the VDWS corresponds to an enlargement of

*Erank.* Meanwhile, Figure 10 also validates this statement. Figures 10a and 10b illustrate that SD enhances *Erank* values of both FS and AT under PGD20 attacks to enhance feature diversity.

### 6.9 Sensitivity Analysis

We examine the parameter sensitivity of $\gamma$ and $\lambda$ on CIFAR-10 in Figure 11. In Figure 11a, we can observe that the effect of the *shirnk* term is greater than that of the *expand* term since the ratio of the *expand* term is $(1 - \gamma)$. Figure 11b shows that our method is less sensitive to $\lambda$ as the fluctuation is only around 2.5%.

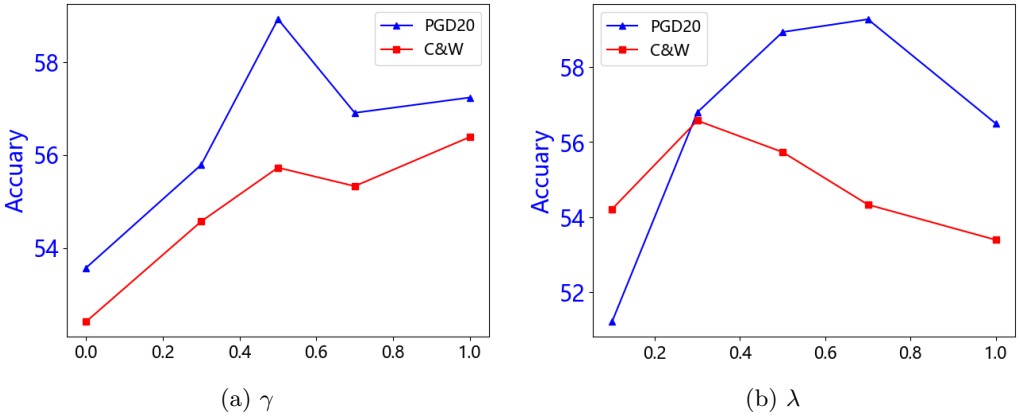

Figure 11: Parameter sensitivity for $\gamma$ and $\lambda$ on CIFAR-10, where all models are trained by AT+SD.

## Conclusion

In this paper, we investigate the robust generalization (RG) problem from the perspective of learning discriminative representation for adversarial examples. Our theoretical and empirical analysis illustrate that reducing the inconsistency of inter-sample relationship maps between clean data and adversarial examples is a feasible approach to alleviate robust overfitting and can be calculated by the proposed Gram matrix difference (GMD). Meanwhile, we provide a theoretical guarantee for RG by introducing a novel and tight error bound based on our GMD. Moreover, to ease the complex optimization of inter-sample relationship maps, we propose a method that expands the volume difference between the entire latent space's linear span and the subspace's linear span, thereby creating a diverging spanned latent space. On the empirical side, we design and implement an adversarial training method, called Subspace Diverging (SD), which alleviates the robust overfitting problem and achieves state-of-the-art performance on multiple benchmarks.

## Acknowledgement

The work was partially supported by the following: National Natural Science Foundation of China under No. 92370119 and 62376113, the WKU Internal (Faculty/Staff) Start-up Research Grant under No. ISRG2024009.

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

## A Theory Proof

We make two assumptions to facilitate the analysis.

**Assumptions:**

• For the neural network $f_\theta(\cdot)$, there exists Lipschitz constant $t$ for loss function $l(\cdot)$ such that for any $x_1, x_2$ satisfying $|l(f_\theta(x_1), y_1) - l(f_\theta(x_2), y_2)| \le t||f_\theta(x_1) - f_\theta(x_2)||_2$.

• For the neural network $f_\theta(\cdot)$, there exists a Lipschitz constant $L$ such that for any $x_1, x_2$ satisfying $||f_\theta(x_1) - f_\theta(x_2)||_2 \le L||x_1 - x_2||_2$.

**Proof of Theorem** 3.1

**Theorem 3.1** *Given the clean data matrix $\boldsymbol{X}_i$ and adversarial data matrix $\boldsymbol{X}_i^{adv}$ that both contain $N_i$ samples of $i$-th class over the training set, the sets of clean data $C_i$ and adversarial data $C_i^{adv}$ of $i$-th class over the underlying data distribution, and the DNN $f_\theta$ that maps data samples to latent features with dimension $r$, if the loss function $l(\cdot)$ of $f_\theta(\cdot)$ is $t$-Lipschitz, and $f_\theta(\cdot)$ is the $L$-Lipschitz, then for any $\sigma > 0$, with the probability at least $1 - \sigma$, we have*

$$\varepsilon_{RGE} \le \varepsilon_{GE} + \frac{tU^2}{N}||\nabla \boldsymbol{T}_d||_2 + tKV^2||\nabla \boldsymbol{T}||_2 + tCHL||\delta||_2 + \sqrt{\frac{2K\ln 2 + 2\ln\left(\frac{1}{\sigma}\right)}{N}},$$

$$where \quad \nabla \boldsymbol{T}_d = \boldsymbol{T}_d^{adv} - \boldsymbol{T}_d, \qquad \nabla \boldsymbol{T} = \boldsymbol{T}^{adv} - \boldsymbol{T},$$

$$U = \frac{1}{N}\sum_{n=1}^{N}||f_\theta(x_n)||_2, \quad V = \frac{1}{K}\sum_{i=1}^{K}||\mathbb{E}[f_\theta(x) \mid x \in C_i]||_2,$$

$$C = 2(N + K^3 + \frac{K^2+1}{2}), \quad H = \sup_x||f_\theta(x)||_2.$$

*$\delta$ is adversarial perturbation and $M$ is the upper bound of the loss function $l(\cdot)$ over the whole underlying data manifold. $\nabla \boldsymbol{T}_d$ and $\nabla \boldsymbol{T}$ denote the Gram matrix difference over the training set and the underlying data distribution respectively,*

$$\nabla \boldsymbol{T}_d = \begin{bmatrix} \left(\boldsymbol{Z}_1^{adv}\right)^\top \boldsymbol{Z}_1^{adv} - \left(\boldsymbol{Z}_1\right)^\top \boldsymbol{Z}_1, & \cdots, & \left(\boldsymbol{Z}_1^{adv}\right)^\top \boldsymbol{Z}_K^{adv} - \left(\boldsymbol{Z}_1\right)^\top \boldsymbol{Z}_K \\ \ddots & \ddots & \ddots \\ \left(\boldsymbol{Z}_K^{adv}\right)^\top \boldsymbol{Z}_1^{adv} - \left(\boldsymbol{Z}_K\right)^\top \boldsymbol{Z}_1, & \cdots, & \left(\boldsymbol{Z}_K^{adv}\right)^\top \boldsymbol{Z}_K^{adv} - \left(\boldsymbol{Z}_K\right)^\top \boldsymbol{Z}_K \end{bmatrix},$$

$$\nabla \boldsymbol{T} = \begin{bmatrix} \left(z_1^{adv}\right)^\top z_1^{adv} - \left(z_1\right)^\top z_1, & \cdots, & \left(z_1^{adv}\right)^\top z_K^{adv} - \left(z_1\right)^\top z_K \\ \ddots & \ddots & \ddots \\ \left(z_K^{adv}\right)^\top z_1^{adv} - \left(z_K\right)^\top z_1, & \cdots, & \left(z_K^{adv}\right)^\top z_K^{adv} - \left(z_K\right)^\top z_K \end{bmatrix},$$

$$where \quad \boldsymbol{Z}_i = \frac{f_\theta(\boldsymbol{X}_i)}{\|f_\theta(\boldsymbol{X}_i)\|_{2,col}}, \quad \boldsymbol{Z}_i^{adv} = \frac{f_\theta\left(\boldsymbol{X}_i^{adv}\right)}{\|f_\theta\left(\boldsymbol{X}_i^{adv}\right)\|_{2,col}}, \quad \boldsymbol{Z}_i, \boldsymbol{Z}_i^{adv} \in \mathbb{R}^{r \times N_i},$$

$$z_i = \frac{\mathbb{E}[f_\theta(x) \mid x \in C_i]}{\|\mathbb{E}[f_\theta(x) \mid x \in C_i]\|_2}, \quad z_i^{adv} = \frac{\mathbb{E}\left[f_\theta\left(x^{adv}\right) \mid x^{adv} \in C_i^{adv}\right]}{\|\mathbb{E}\left[f_\theta(x^{adv}) \mid x \in C_i^{adv}\right]\|_2}, \quad z_i, z_i^{adv} \in \mathbb{R}^r.$$
(12)

*$\|\cdot\|_{2,col}$ represents the calculation of the Euclidean norm of column vectors in the matrix.*

*proof* : Let $(|N_1|, \cdots, |N_K|)$ is an IID multinomial random variable with parameters $N$. $N$ and $K$ denote the total number of training samples and the total number of classes respectively. With the probability distribution set for the underlying classes $\{\mu(C_1), \mu(C_2), \ldots, \mu(C_i), \ldots \mu(C_K)\}$, following Breteganolle-Huber-Carel inequality (Proposition A6.6 of (Wellner et al., 2013)):

$$Pr\{\sum_{i=K}^{K}|\frac{|N_i|}{N} - \mu(C_i)| \ge \lambda\} \le 2^K exp(\frac{-2N\lambda^2}{2}).$$

With the probability at least $1 - \delta$, we can get:

$$\sum_{i=1}^{K} \left| \frac{|N_i|}{N} - \mu(C_i) \right| \leq \sqrt{\frac{2K \ln 2 + 2 \ln \left(\frac{1}{\sigma}\right)}{N}}.$$

$$
\begin{aligned}
\varepsilon_{RGE} &= \left| \mathbb{E}\left[ l\left(f_\theta(x^{adv}), y\right) \mid x \in X \right] - \frac{1}{N} \sum_{n=1}^{N} l\left(f_\theta(x_n^{adv}), y_n\right) \right| \\
&= \left| \sum_{i=1}^{K} \mathbb{E}\left[ l\left(f_\theta(x^{adv}), y\right) \mid x \in C_i \right] \mu(C_i) - \frac{1}{N} \sum_{n=1}^{N} l\left(f_\theta(x_n^{adv}), y_n\right) \right| \\
&= \left| \sum_{i=1}^{K} \left[ \mathbb{E}\left[ l\left(f_\theta(x^{adv}), y\right) \mid x \in C_i \right] + \mathbb{E}\left[ l\left(f_\theta(x), y\right) \mid x \in C_i \right] - \mathbb{E}\left[ l\left(f_\theta(x), y\right) \mid x \in C_i \right] \right] \mu(C_i) - \right. \\
&\qquad \left. \frac{1}{N} \sum_{n=1}^{N} \left[ l\left(f_\theta(x_n^{adv}), y_n\right) \quad - l\left(f_\theta(x_n), y_n\right) + l\left(f_\theta(x_n), y_n\right) \right] \right| \\
&\leq \left| \sum_{i=1}^{K} \mathbb{E}\left[ l\left(f_\theta(x), y\right) \mid x \in C_i \right] \mu(C_i) - \frac{1}{N} \sum_{n=1}^{N} l\left(f_\theta(x_n), y_n\right) \right| + \left| \sum_{i=1}^{K} \left[ \mathbb{E}\left[ l\left(f_\theta(x^{adv}), y\right) \mid x \in C_i \right] - \right. \right. \\
&\qquad \left. \left. \mathbb{E}\left[ l\left(f_\theta(x), y\right) \mid x \in C_i \right] \mu(C_i) \quad - \frac{1}{N} \sum_{n=1}^{N} \left[ l\left(f_\theta(x_n^{adv}), y_n\right) - l\left(f_\theta(x_n), y_n\right) \right] \right| . \right.
\end{aligned}
$$

$$
\begin{aligned}
&= \varepsilon_{GE} + \left| \sum_{i=1}^{K} \left[ \mathbb{E}\left[ l\left(f_\theta(x^{adv}), y\right) \mid x \in C_i \right] - \mathbb{E}\left[ l\left(f_\theta(x), y\right) \mid x \in C_i \right] \right] \mu(C_i) - \frac{1}{N} \sum_{n=1}^{N} \left[ l\left(f_\theta(x_n^{adv}), y_n\right) - l\left(f_\theta(x_n), y_n\right) \right] \right. \\
&\quad + \sum_{i=1}^{K} \left[ \mathbb{E}\left[ l\left(f_\theta(x^{adv}), y\right) \mid x \in C_i \right] - \mathbb{E}\left[ l\left(f_\theta(x), y\right) \mid x \in C_i \right] \right] \frac{|N_i|}{N} \\
&\quad \left. - \sum_{i=1}^{K} \left[ \mathbb{E}\left[ l\left(f_\theta(x), y\right) \mid x \in C_i \right] - \mathbb{E}\left[ l\left(f_\theta(x), y\right) \mid x \in C_i \right] \right] \frac{|N_i|}{N} \right| \\
&\leq \varepsilon_{GE} + \left| \sum_{i=1}^{K} \left[ \mathbb{E}\left[ l\left(f_\theta(x^{adv}), y\right) \mid x \in C_i \right] - \mathbb{E}\left[ l\left(f_\theta(x), y\right) \mid x \in C_i \right] \right] \mu(C_i) - \sum_{i=1}^{K} \left[ \mathbb{E}\left[ l\left(f_\theta(x^{adv}), y\right) \mid x \in C_i \right] - \right. \right. \\
&\quad \left. \left. \mathbb{E}\left[ l\left(f_\theta(x), y\right) \mid x \in C_i \right] \right] \frac{|N_i|}{N} \right| \\
&\quad + \left| \frac{1}{N} \sum_{n=1}^{N} \left[ l\left(f_\theta(x_n^{adv}), y_n\right) - l\left(f_\theta(x_n), y_n\right) \right] - \frac{|N_i|}{N} \sum_{i=1}^{K} \left[ \mathbb{E}\left[ l\left(f_\theta(x^{adv}), y\right) - l\left(f_\theta(x), y\right) \mid x \in C_i \right] \right] \right| .
\end{aligned}
$$

We define the loss upper bound $\displaystyle \sup_{x^{adv} \in X^{adv}} f(x^{adv})$ as $M$, where $X^{adv}$ represents the distribution of adversarial examples corresponding to the underlying data distribution. Therefore, we can get

$$
\begin{aligned}
&\leq \varepsilon_{GE} + \left| \sum_{i=1}^{K} \mathbb{E}\left[ l\left(f_\theta(x^{adv}), y\right) - l\left(f_\theta(x), y\right) \mid x \in C_i \right] \frac{|N_i|}{N} - \frac{1}{N} \sum_{n=1}^{N} \left[ l\left(f_\theta(x_n^{adv}), y_n\right) - l\left(f_\theta(x_n), y_n\right) \right] \right| \\
&\quad + M \cdot \left| \sum_{i=1}^{K} \left( \mu(C_i) - \frac{|N_i|}{N} \right) \right| .
\end{aligned}
$$

Given that $\{|N_i|\}_{i=1}^K$ is a multinomial random variable with parameter $N$, we apply the Bretagnolle-Huber-Carel inequality to obtain:

$$\leq \varepsilon_{GE} + \left| \sum_{i=1}^K \mathbb{E}\left[ l\left(f_\theta(x^{adv}), y\right) - l\left(f_\theta(x), y\right) \mid x \in C_i \right] \frac{|N_i|}{N} - \frac{1}{N}\sum_{n=1}^N \left[ l\left(f_\theta(x_n^{adv}), y_n\right) - l(f_\theta(x_n), y_n) \right] \right|$$

$$+ M \cdot \sqrt{\frac{2K\ln 2 + 2\ln\left(\frac{1}{\sigma}\right)}{N}}$$

$$\leq \varepsilon_{GE} + \frac{t}{N} * \left| \sum_{n=1}^N \left[ ||f_\theta(x_n^{adv}) - f_\theta(x_n)||_2 \right] + |N_i|\sum_{i=1}^K \mathbb{E}\left[ ||(f_\theta(x^{adv}) - f_\theta(x))||_2 \mid x \in C_i \right] \right| + M \cdot \sqrt{\frac{2K\ln 2 + 2\ln\left(\frac{1}{\sigma}\right)}{N}}$$

$$\leq \varepsilon_{GE} + \frac{t}{N} * \sum_{n=1}^N ||f_\theta(x_n^{adv}) - f_\theta(x_n)||_2 + \sum_{i=1}^K \frac{t|N_i|}{N} \mathbb{E}\left[ ||(f_\theta(x^{adv}) - f_\theta(x))||_2 \mid x \in C_i \right] + M \cdot \sqrt{\frac{2K\ln 2 + 2\ln\left(\frac{1}{\sigma}\right)}{N}}.$$

Let $f_\theta(\boldsymbol{X}_d) \in \mathbb{R}^{r \times N}$ and $f_\theta(\boldsymbol{X}_u) \in \mathbb{R}^{r \times K}$ denote the training and underlying feature matrix respectively, where $f_\theta(\boldsymbol{X}_u) = [\mathbb{E}[f(x)] \mid x \in C_i]_{i=1}^K$; and $f_\theta\left(\boldsymbol{X}_d^{adv}\right) \in \mathbb{R}^{r \times N}$ and $f_\theta\left(\boldsymbol{X}_u^{adv}\right) \in \mathbb{R}^{r \times K}$ denote the corresponding adversarial feature sets. Then, since $|N_i| \leq N$, we can deduce that

$$\leq \varepsilon_{GE} + \frac{t}{N}||f_\theta\left(\boldsymbol{X}_d^{adv}\right) - f_\theta\left(\boldsymbol{X}_d\right)||_2 + Kt||\mathbb{E}\left[f_\theta(\boldsymbol{X}_u^{adv})\right] - \mathbb{E}\left[f_\theta(\boldsymbol{X}_u)\right]||_2 + M \cdot \sqrt{\frac{2K\ln 2 + 2\ln\left(\frac{1}{\sigma}\right)}{N}},$$

$$= \varepsilon_{GE} + \frac{t}{N}\left[ ||f_\theta\left(\boldsymbol{X}_d^{adv}\right) - f_\theta\left(\boldsymbol{X}_d\right)||_2 + ||\left[f_\theta\left(\boldsymbol{X}_d^{adv}\right)\right]^2 - \left[f_\theta\left(\boldsymbol{X}_d\right)\right]^2||_2 - ||\left[f_\theta\left(\boldsymbol{X}_d^{adv}\right)\right]^2 - \left[f_\theta\left(\boldsymbol{X}_d\right)\right]^2||_2 \right]$$

$$+ Kt\left[ ||\mathbb{E}\left[f_\theta(\boldsymbol{X}_u^{adv})\right] - \mathbb{E}\left[f_\theta(\boldsymbol{X}_u)\right]||_2 + ||\mathbb{E}\left[\left[f_\theta(\boldsymbol{X}_u^{adv})\right]\right]^2 - \mathbb{E}\left[\left[f_\theta(\boldsymbol{X}_u)\right]^2\right]||_2 - ||\mathbb{E}\left[\left[f_\theta(\boldsymbol{X}_u^{adv})\right]^2\right] - \mathbb{E}\left[\left[f_\theta(\boldsymbol{X}_u)\right]^2\right]||_2 \right]$$

$$+ M \cdot \sqrt{\frac{2K\ln 2 + 2\ln\left(\frac{1}{\sigma}\right)}{N}}$$

$$\leq \varepsilon_{GE} + \frac{t}{N}||\left[f_\theta(\boldsymbol{X}_D^{adv})\right]^2 - \left[f_\theta(\boldsymbol{X}_D)\right]^2||_2 + Kt||\mathbb{E}\left[f_\theta(\boldsymbol{X}_u^{adv})\right]^2 - \mathbb{E}\left[f_\theta(\boldsymbol{X}_u)\right]^2||_2$$

$$+ \frac{t}{N}\left[ ||f_\theta(X_D^{adv}) - f_\theta(X_D)||_2 \left( ||f_\theta(X_D^{adv}) + f_\theta(X_D)||_2 + 1 \right) \right]$$

$$+ Kt\left[ ||\mathbb{E}\left[f_\theta(X_u^{adv})\right] - \mathbb{E}\left[f_\theta(X_u)\right]||_2 \left( ||\mathbb{E}\left[f_\theta(X_u^{adv})\right] + \mathbb{E}\left[f_\theta(X_u)\right]||_2 + 1 \right) \right]$$

$$+ M \cdot \sqrt{\frac{2K\ln 2 + 2\ln\left(\frac{1}{\sigma}\right)}{N}}$$

$H$ is defined as $\sup_x ||f(x)||_2$, which, together with the assumption, we can gain:

$$\leq \varepsilon_{GE} + \frac{t}{N}||\left[f_\theta(\boldsymbol{X}_D^{adv})\right]^2 - \left[f_\theta(\boldsymbol{X}_D)\right]^2||_2 + Kt||\mathbb{E}\left[f_\theta(\boldsymbol{X}_u^{adv})\right]^2 - \mathbb{E}\left[f_\theta(\boldsymbol{X}_u)\right]^2||_2 + 2t(N + K^3 + \frac{K^2+1}{2})HL||\delta||_2$$

$$+ M \cdot \sqrt{\frac{2K\ln 2 + 2\ln\left(\frac{1}{\sigma}\right)}{N}}$$

$$= \varepsilon_{GE} + \frac{tU^2}{N}||\left(\boldsymbol{T}_d^{adv} - \boldsymbol{T}_d\right)||_2 + tKV^2||\left(\boldsymbol{T}^{adv} - \boldsymbol{T}\right)||_2 + 2t(N + K^3 + \frac{K^2+1}{2})HL||\delta||_2 + M \cdot \sqrt{\frac{2K\ln 2 + 2\ln\left(\frac{1}{\sigma}\right)}{N}}$$

$$= \varepsilon_{GE} + \frac{tU^2}{N}||\nabla\boldsymbol{T}_d||_2 + tKV^2||\nabla\boldsymbol{T}||_2 + 2t(N + K^3 + \frac{K^2+1}{2})HL||\delta||_2 + M \cdot \sqrt{\frac{2K\ln 2 + 2\ln\left(\frac{1}{\sigma}\right)}{N}}.$$

**Proof of Theorem** 3.2

**Theorem 3.2** *Let $\boldsymbol{S}$ be the span of the latent space, encompassing all subspaces $\{\boldsymbol{S}_i\}_{i=1}^K \subset \boldsymbol{S}$. Let $\boldsymbol{I}$ denote the identity matrix. Then,*

$$
\begin{aligned}
If \quad &\text{Vol}(\boldsymbol{S}) \quad \text{is maximized, all subspaces } \boldsymbol{S}_i \subset \boldsymbol{S} \text{ are mutually independent,} \\
If \quad &\text{Vol}(\boldsymbol{I} + \boldsymbol{S}_i) \quad \text{is minimized, the vectors in } \boldsymbol{S}_i \text{ remain consistent.}
\end{aligned}
\tag{13}
$$

**Proof:**

To establish the proof, we initially introduce the $\log \det(\cdot)$ function, leveraging its concavity as detailed in (Boyd & Vandenberghe, 2004).Since $\log \det(\boldsymbol{I} + \boldsymbol{S}) = \log \det(\boldsymbol{I} + \boldsymbol{Z}^\top \boldsymbol{Z})$ holds for feature representations, this equivalence enables the following results:

$$
\max_{\boldsymbol{Z}} \ \log \det(\boldsymbol{S}) \iff \max_{\boldsymbol{Z}} \ \log \det(\boldsymbol{I} + \boldsymbol{Z}^\top \boldsymbol{Z}).
\tag{14}
$$

$\boldsymbol{Z}$ denote the normalized feature matrix of $f_\theta(\boldsymbol{X})$, where $\boldsymbol{Z} = \{\boldsymbol{Z}_i\}_{i=1}^K$. $\boldsymbol{Z}_i$ represents the feature representation matrix for the $i$-th class.

First, we prove that $\max\limits_{Z} \ \log \det(\boldsymbol{S})$ can achieve all subspace independent.

We define the matrix $\boldsymbol{D}$ as:

$$
\boldsymbol{D} = \boldsymbol{I} + \boldsymbol{Z}^\top \boldsymbol{Z}.
\tag{15}
$$

The expression $\boldsymbol{D}$ can be written as:

$$
\boldsymbol{D} = I + \begin{bmatrix}
\boldsymbol{Z}_1^\top \boldsymbol{Z}_1 & \boldsymbol{Z}_1^\top \boldsymbol{Z}_2 & \cdots & \boldsymbol{Z}_1^\top \boldsymbol{Z}_k \\
\boldsymbol{Z}_2^\top \boldsymbol{Z}_1 & \boldsymbol{Z}_2^\top \boldsymbol{Z}_2 & \cdots & \boldsymbol{Z}_2^\top \boldsymbol{Z}_k \\
\vdots & \vdots & \ddots & \vdots \\
\boldsymbol{Z}_k^\top \boldsymbol{Z}_1 & \boldsymbol{Z}_k^\top \boldsymbol{Z}_2 & \cdots & \boldsymbol{Z}_k^\top \boldsymbol{Z}_k
\end{bmatrix}.
$$

Furthermore, we define $\boldsymbol{D^*}$ as:

$$
\boldsymbol{D^*} = I + \begin{bmatrix}
\boldsymbol{Z}_1^\top \boldsymbol{Z}_1 & 0 & \cdots & 0 \\
0 & \boldsymbol{Z}_2^\top \boldsymbol{Z}_2 & \cdots & 0 \\
\vdots & \vdots & \ddots & \vdots \\
0 & 0 & \cdots & \boldsymbol{Z}_k^\top \boldsymbol{Z}_k
\end{bmatrix}.
$$

Consider $\boldsymbol{D}^*$, in which all subspaces are independent. If the volume of $\boldsymbol{D}^*$ (i.e., $\det(\boldsymbol{D}^*)$) is greater than that of $\boldsymbol{D}$ (i.e., $\det(\boldsymbol{D})$), then our theorem can be established. We assume that $\boldsymbol{D}$ is a non-singular matrix and denote $\boldsymbol{D}^\top$ as the transpose matrix of $\boldsymbol{D}$. Because the strict concavity of $\log \det(\cdot)$, we can get the relationship as follow:

$$
\log \det(\boldsymbol{D}) - \log \det(\boldsymbol{D}^*) \leq \ \langle \nabla \log \det(\boldsymbol{D}^*), \boldsymbol{D} - \boldsymbol{D}^* \rangle.
\tag{16}
$$

From (Boyd & Vandenberghe, 2004), we can get that $\nabla \log \det(\boldsymbol{D}^*) = \boldsymbol{D}^{*-1}$. In addition, due to $\boldsymbol{D}^{*-1} = \left(\boldsymbol{D}^{*-1}\right)^\top$, the right hand side (RHS) of Equation 16 can be shown as:

$$
\begin{aligned}
&= \langle \left(\boldsymbol{D}^{*-1}\right), (\boldsymbol{D} - \boldsymbol{D}^*) \rangle \\
&= \langle \left(\boldsymbol{D}^{*-1}\right), \boldsymbol{D} \rangle - \langle \left(\boldsymbol{D}^{*-1}\right), \boldsymbol{D}^* \rangle
\end{aligned}
$$

We set the matrix $\left(\boldsymbol{D}^{*-1}\right)\boldsymbol{D}$ as $\boldsymbol{G}$

$$
\begin{aligned}
&= \operatorname{tr}(\boldsymbol{G}) - \operatorname{tr}(\boldsymbol{I}) \\
&= N - N \\
&= 0.
\end{aligned}
$$

The matrix $\boldsymbol{G}$ can be represented as follows:

$$
\boldsymbol{G} = \begin{bmatrix} \left(I + \boldsymbol{Z}_1^\top \boldsymbol{Z}_1\right)^{-1} \left(I + \boldsymbol{Z}_1^\top \boldsymbol{Z}_1\right) & \cdots & \left(I + \boldsymbol{Z}_1^\top \boldsymbol{Z}_1\right)^{-1} \left(I + \boldsymbol{Z}_1^\top \boldsymbol{Z}_k\right) \\ \vdots & \ddots & \vdots \\ \left(I + \boldsymbol{Z}_k^\top \boldsymbol{Z}_1\right)^{-1} \left(I + \boldsymbol{Z}_k^\top \boldsymbol{Z}_1\right) & \cdots & \left(I + \boldsymbol{Z}_k^\top \boldsymbol{Z}_k\right)^{-1} \left(I + \boldsymbol{Z}_k^\top \boldsymbol{Z}_k\right) \end{bmatrix} = \begin{bmatrix} 1 & \cdots & \emptyset \\ \vdots & \ddots & \vdots \\ \emptyset & \cdots & 1 \end{bmatrix}
$$

where $\emptyset$ are the irrelevant numbers with the process of calculating trace. Thus, when the volume of the latent span $\det(\boldsymbol{D}^*)$ reaches its maximum value, each subspace is independent of each other.

Then we proceed to prove that if $\operatorname{Vol}(\boldsymbol{I} + \boldsymbol{S}_i)$ is minimized, then the vectors within $\boldsymbol{S}_i$ remain consistent. The same theory as above, the relationship is also satisfied:

$$
\min_{\boldsymbol{Z}_i} \ \log\det(\boldsymbol{I} + \boldsymbol{S}_i) \iff \min_{\boldsymbol{Z}_i} \log\det(\boldsymbol{I} + \boldsymbol{Z}_i^\top \boldsymbol{Z}_i), \tag{17}
$$

where $\boldsymbol{Z}_i \in \mathbb{R}^{r \times T}$ is $i$-th class feature matrix, which includes T samples, and $\boldsymbol{Z}_i = [z_{i_t}]_{t=1}^{\mathrm{T}}$. In addition, $\log\det(\boldsymbol{I} + \boldsymbol{Z}_i^\top \boldsymbol{Z}_i)$ and $\log\det(\boldsymbol{Z}_i^\top \boldsymbol{Z}_i)$ share the same trends. The matrix $\boldsymbol{I} + \boldsymbol{Z}_i^\top \boldsymbol{Z}_i$ can be defined as matrix $\boldsymbol{Q}_i$:

$$
\boldsymbol{Q}_i = \boldsymbol{I} + \begin{bmatrix} \langle z_{i_1}, z_{i_1}\rangle & \cdots & \langle z_{i_1}, z_{i_{\mathrm{T}}}\rangle \\ \vdots & \ddots & \vdots \\ \langle z_{i_{\mathrm{T}}}, z_{i_1}\rangle & \cdots & \langle z_{i_{\mathrm{T}}}, z_{i_{\mathrm{T}}}\rangle \end{bmatrix}. \tag{18}
$$

Consider the matrix $\boldsymbol{Z}_i^\top \boldsymbol{Z}_i$. Since its diagonal elements are equal to 1 and the off-diagonal elements are less than or equal to 1, it follows that $\det(\boldsymbol{Z}_i^\top \boldsymbol{Z}_i) \geq 0$. For any vector $a, b \in \boldsymbol{Z}_i$, and $\langle a, b\rangle = 1$, $\det(\boldsymbol{Z}i^\top \boldsymbol{Z}i)$ will be close to minimum value of 0. Correspondingly, $\log\det(\boldsymbol{Q}_i)$ will also reach its minimization value.

Our proof has been finished.

## A.1 Indirect Comparison between SIC and GMD

For indirect comparison, we illustrate that our GMD is equivalent to a lower bound (lowest value) of SCR term. As a result, using our GMD to upper bound the robust generation error will result in a tighter upper bound than using SCR. Specifically, the lower bound of the SCR term is expressed as:

$$
\begin{aligned}
SCR &= \min_\theta \frac{t}{N} \sum_{i=1}^{K} \sum_{v \in \hat{N}_i} \| f_\theta(\boldsymbol{x}_v^{adv}) - f_\theta(\boldsymbol{x}_v) - \mathbb{E}\left[f_\theta(\boldsymbol{x}^{adv}) - f_\theta(\boldsymbol{x}) \mid \boldsymbol{x} \in C_i\right] \|_2^2 \\
&\geq \min_\theta \frac{t}{N} \| \sum_{i=1}^{K} \sum_{v \in \hat{N}_i} \left(f_\theta(\boldsymbol{x}_v^{adv}) - f_\theta(\boldsymbol{x}_v)\right) - N \sum_{i=1}^{K} \mathbb{E}\left[f_\theta(\boldsymbol{x}^{adv}) - f_\theta(\boldsymbol{x}) \mid \boldsymbol{x} \in C_i\right] \|_2^2 \\
&\geq \min_\theta \frac{t}{N} \left| \| \sum_{i=1}^{K} \sum_{v \in \hat{N}_i} \left(f_\theta(\boldsymbol{x}_v^{adv}) - f_\theta(\boldsymbol{x}_v)\right) \|_2^2 - N \sum_{i=1}^{K} \| \mathbb{E}\left[f_\theta(\boldsymbol{x}^{adv}) - f_\theta(\boldsymbol{x}) \mid \boldsymbol{x} \in C_i\right] \|_2^2 \right| \\
&= \min_\theta \frac{t}{N} \left| \| f(\boldsymbol{X}_D^{adv}) - f(\boldsymbol{X}_D) \|_2^2 - \| f(\boldsymbol{X}^{adv}) - f(\boldsymbol{X}) \|_2^2 \right|.
\end{aligned}
$$

When $f(\boldsymbol{X}^{adv}) \to f(\boldsymbol{X})$ and $f(\boldsymbol{X}_D^{adv}) \to f(\boldsymbol{X}_D)$, the resulting term,

$$\min_\theta \frac{t}{N} \left| \|f(\boldsymbol{X}_D^{adv}) - f(\boldsymbol{X}_D)\|_2^2 - \|f(\boldsymbol{X}^{adv}) - f(\boldsymbol{X})\|_2^2 \right|,$$

provides a more constrained optimization compared to SCR. This optimization encourages not only intra-class consistency, as is the case with SCR, but also improves inter-class separation, which is essential for better generalization and robustness against adversarial attacks.

For our GMD term,

$$\min_\theta \left[ ||\nabla \boldsymbol{T}_d||_2 + ||\nabla \boldsymbol{T}||_2 \right] = \min_\theta \Big[ || \left[ f_\theta(\boldsymbol{X}_D^{adv}) - f_\theta(\boldsymbol{X}_D) \right] \left[ f_\theta(\boldsymbol{X}_D^{adv}) + f_\theta(\boldsymbol{X}_D) \right] ||_2$$
$$+ || \left[ f_\theta(\boldsymbol{X}^{adv}) - f_\theta(\boldsymbol{X}) \right] \left[ f_\theta(\boldsymbol{X}^{adv}) + f_\theta(\boldsymbol{X}) \right] ||_2 \Big].$$

it also encourages both $f(\boldsymbol{X}_D^{adv}) \to f(\boldsymbol{X}_D)$ and $f(\boldsymbol{X}^{adv}) \to f(\boldsymbol{X})$.

Based on the above analysis, the optimization of GMD is a tighter and more constrained approach compared to SCR.

