# OpenReview forum: "Improving Robust Generalization with Diverging Spanned Latent Space"
_TMLR — Accepted by TMLR_

### Review · Reviewer_aW6S · 2024-09-06

**Summary Of Contributions:**

The paper uses the similarity in feature space to define and optimize a novel subspace divergence metric. They motivate their approach with theoretical arguments based on an inequality on the robust generalization gap involving differences in the class gram matrices under adversarial attacks. The paper proposes an algorithm to implement a loss term to minimize the robust generalization gap and gives numerical examples of its efficacy.

**Audience:**

Yes

**Broader Impact Concerns:**

I have no concerns.

**Claims And Evidence:**

No

**Requested Changes:**

## Rethink the methodology of the experimental section
- **[C1]** Focus on fewer experiments and conduct them in a sound way. I.e. consistent hyperparameter tuning, statistically significant results.
- **[C2]** The experimental section is hard to read and IMO unnecessarily wordy you can save some space and make your contribution clearer by reworking it. The experimental section right now overshadows and diminishes the conceptual results. Take, i.e., 6.2 paragraph 3 is hard to read as there is an abundance of abbreviations, and as far as I understand, there is no insight, but rather parts of table 2 spelled out in the text. A table is a good medium for comparing numbers, the text should focus on the main result and its interpretation.
- **[C3]** While the authors mention the computational intractability of a regularizer and suggest a batch approximation, the actual computational time and memory complexities are not analyzed. Based on the selected hyperparameters it also seems like the method needs rather large batch sizes. These practical issues should be addressed.
## Minor
- Phrases like p.13 "These results prove that SD can achieve better robust accuracy compared to SCR." Do not help in interpreting the result and are vague: can they achieve that with specific datasets or attacks, on average, ...
- 6.5.2 "increases the inter-class similarity" decreases?
- In the proof section, assumptions are made on the fly; they should be stated before or at the beginning of the theorem.
- p.5 in the inequality for $\epsilon_{RGE}$ there is a $\sigma$ that should probably be a $\delta$.

**Strengths And Weaknesses:**

## Strengths
- **[S1]** The idea of intersample and intrasample similarities is intuitive.
- **[S2]** The approach is well-motivated.
- **[S3]** The paper gives experiments that analyze the properties of their algorithm (diversity of features, sensitivity of parameters). In my opinion validating and analyzing the proposed algorithm is more insightful than another empirical comparison.

## Weaknesses
- **[W1]** On p.6 neglecting the GMD on the data distribution is motivated in a very heuristic way, I was not fully convinced. It would strengthen your argument if you provided some mathematical explanation or made the assumption more explicit and formalized.
- **[W2]** The experimental section is weak. While the authors provide lots of comparison methods and apply their procedure to 3 computer vision datasets, the methodology makes the results questionable.
  - [W2.1] There is no hint of hyperparameter selection and as the authors analyze in 6.5.4 improved results could well be due to more extensive hyperparameter selection
  - [W2.2] The experiments seem to not be repeated, and as such, there are no standard deviations.
  - [W2.3] No meta-analysis was done in an attempt to compare the methods rigorously over the set of dataset variations and attacks that were tested.
- **[W3]** The inequality on the robust generalization gap is sometimes called "tight"; as far as I understand the result, this is not proven.
- **[W4]** The overall paper is not polished: There are typos and stylistic errors, ie, missing axis labels, one or two-sentence sections, tables not complying with the margins, redundant descriptions of experiments, and incorrect highlighting in tables.

---

> ### Author Response · Authors · 2024-10-20
>
> ### W1: Neglecting the GMD on the Data Distribution
>
> Thank you for your insightful feedback.
>
> We did not ignore the GMD on the underlying data distribution but assumed that this term will change with the GMD on the training data distribution since the underlying data distribution is unavailable in the real world. In other words, $ ||\nabla \boldsymbol{T}_{d}||_2$  decreases with
> $\left|\left|\nabla \boldsymbol{T}_u\right|\right| _2$, i.e., $\min _\theta||  \nabla \boldsymbol{T}_d ||_2 \rightarrow \min _\theta||\nabla \boldsymbol{T}||_2$
>
>
>
> This assumption is reasonable because it follows a basic assumption of machine learning methods: statistically, improving the feature representation on the available training set is essential and can often be assumed to generalize to improving the learned feature representation on the unknown underlying data distribution. It is important to note that, in order to bridge the gap between the unknown underlying data and the available training data, this is a popular approach in prior works on adversarial robustness and representation learning [@bui2020improving; @li2019improving; 3], which lend support to our approach.
>
> Meanwhile, the GMD is determined by both inter-class distance and intra-class distance. Our goal is to increase inter-class distance while simultaneously decreasing intra-class distance on the training set, which fosters more discriminative feature representations. Such adjustments on the training set are likely to reflect similar trends in the entire data distribution.
>
> ### W2: Experiment concerns
>
> #### W2.1: Hyper-parameter selection
> As for your concern "**improved results could well be due to more extensive hyper-parameter selection**," we understand your concern but we may not agree with you. Please allow us to clarify this as follows.
>
> In response to your concern about "**no hint about hyper-parameter selection**," we actually used grid search to tune parameters $\gamma$, $\lambda$, and $\omega$.
> In Table 1 of our main paper, $d$ and $k$ are normalization parameters that are measured during training as shown in Algorithm 1 rather than selected; $\gamma$, $\omega$, and $\lambda$ work as internal parameters that are selected by grid search and are not related to the training methods of baselines. Regarding the training epochs and batch size, almost all comparison methods, such as RAT, LAS, and SCR, are different. Moreover, it is common for these to vary depending on the baseline and dataset [3; 4; 5]. Other parameters, such as learning rate and weight decay, are kept consistent with the baselines for a fair comparison.
>
> In short, we basically follow the commonly used hyper-parameter selection strategy in the machine learning literature that ensures a fair comparison among different methods. We believe our produced improvements are consistent, independent of the baselines, and do not rely on over hyper-parameter tuning for baselines.
>
> #### W2.2: Standard deviations
> We have added standard deviations calculated over five runs with different seeds in the experiments in the main paper.
>
> #### W2.3: Meta-analysis
> We provide a meta-analysis in Section 6.7 of the main paper. To rigorously quantify the differences in performance, we employed two statistical measures: Cohen’s effect size and the t-test.
>
> [1] Li, P., Yi, J., Zhou, B., & Zhang, L. (2019). Improving the robustness of deep neural networks via adversarial training with triplet loss.
>
> [2] Bui, A., Le, T., Zhao, H., Montague, P., deVel, O., Abraham, T., & Phung, D. (2020, August). Improving adversarial robustness by enforcing local and global compactness. In European Conference on Computer Vision (pp. 209-223). Cham: Springer International Publishing.
>
> [3] Jin, G., Yi, X., Wu, D., Mu, R., & Huang, X. (2023). Randomized adversarial training via taylor expansion. In Proceedings of the IEEE/CVF Conference on Computer Vision and Pattern Recognition (pp. 16447-16457).
>
> [4] Qian, Z., Zhang, S., Huang, K., Wang, Q., Yi, X., Gu, B., & Xiong, H. (2024). Perturbation diversity certificates robust generalization. Neural Networks, 172, 106117.
>
> [5] Zhang, S., Qian, Z., Huang, K., Wang, Q. F., Gu, B., Xiong, H., & Yi, X. (2024). Inter-feature Relationship Certifies Robust Generalization of Adversarial Training. International Journal of Computer Vision, 1-17.

---

> ### Author Response · Authors · 2024-10-20
>
> ### W4: Errors and typos in the paper
>
> Thank you for pointing this out. We have thoroughly checked the paper and corrected many errors and typos accordingly.
>
> ### C1: Consistent hyper-parameter tuning
>
> Thank you for your feedback. After careful consideration and analysis, we believe that adjusting the ratio between the adversarial loss and the regularization term (SD loss) to achieve better performance is a reasonable and common practice. As stated in W2.1, our hyper-parameters ($d$, $k$) are measured, while other hyper-parameters related to the SD regularization term are internal and independent of the baselines. We have kept other parameters, such as learning rate and weight decay, consistent with the baselines. Therefore, we believe that it is reasonable for the internal hyper-parameters of SD to vary across different datasets. Nonetheless, we thank you for your suggestion, and we will further clarify and improve the algorithm based on your feedback.
>
> ### C2: Rewrite experiment
>
> We have reworked our experimental section and provided a more concise presentation.
>
> ### C3: Batch approximation
>
> Thank you for your comment.
> To achieve the learning objective, our method leverages batch approximation, which is a standard practice in the research community. While it is true that larger batch sizes may better approximate the results, our SD still works effectively using batch sizes similar to the baselines.
>
> ### M1: Concern about robust accuracy compared to SCR
>
> Thank you for your valuable advice, and we have tried our best to polish our descriptions in the paper.
> As observed in the results, our SD achieves better robust accuracy than SCR in most experiments.
> In Tables 2 to 7, our SD consistently improves the baselines and surpasses SCR across different datasets and attack types. Especially in Table 6, since SCR is based on the FS baseline, we conducted a focused comparison between SCR and SD under this baseline. While SD produced three better robust accuracy results than SCR on the CIFAR10 dataset, our SD consistently outperformed SCR across all the other datasets and attack settings.
>
> ### M2 to M4：
>
> We have made the revisions from minor 2 to minor 4 in our polished paper.

---

> ### Author Response · Authors · 2024-10-20
>
> ### W3：a tighter bound
> Thanks for your raising the question which allows us to further clarify our work. Our claim of being tighter is based on a comparison with SCR. Directly comparing GMD with SCR is challenging due to their distinct optimization objectives and forms. Therefore, we considered conducting an indirect comparison between the SCR and SD.
>
>
> For indirect comparison, we illustrate that our GMD is equivalent to a lower bound (lowest value) of SCR term.
> As a result, using our GMD to upper bound the robust generation error will results in a more tighter upper bound than using SCR.
>
>
> We have put our proof details in our revision (Appendix A.1).

---

> ### Author Response · Authors · 2024-10-21
> **More proof details**
>
> Dear Reviewer,
>
> Thank you for your insightful comments. Regarding your question about **W3: "tighter bound"
> , due to Markdown rendering issues in OpenReview, we have included the detailed derivation and explanation in the supplementary material to ensure that the formulas and proofs are presented correctly. We hope this supplementary content provides a clearer response to your concerns.**
>
> Thank you again for your valuable feedback!
>
> Best regards,

---

### Review · Reviewer_thck · 2024-09-13

**Summary Of Contributions:**

In this paper, an algorithm to enhance robust generalization is provided. The authors introduced the Gram matrix difference as a regularization term to tighten the upper bound on the error. The authors utilized the Subspace Diverging method for adversarial training to extend the difference between the current span of the potential space and the subspace. The authors then demonstrate experimentally that their method improves relative to the baseline.

**Audience:**

Yes

**Claims And Evidence:**

Yes

**Requested Changes:**

The experimental part of the paper is not sufficient and some of the results are not advanced enough.

1. The limited effectiveness of the authors' method under AA casts doubt on the effectiveness of this method under complex attacks. It is hoped that the authors can conduct experiments for more complex attacks.

2. The authors need more ablation experiments to demonstrate the effectiveness of their method at different attack intensities.

3. The authors' design for the experimental hyperparameters in Table 1 is tricky. Hopefully, the authors can explain this in more detail, such as values other than the estimation of d,k. This would prove that the authors' method is useful and not a tricky choice of hyperparameters.

**Strengths And Weaknesses:**

1. The authors give good theoretical reasoning.

2. Excellent visualization makes reasoning about spatial distances convincing.

---

> ### Author Response · Authors · 2024-10-20
>
> ## C1: More complex attack
>
> Thanks for your valuable comments. We further evaluate the robustness of different adversarial training methods against the Maximum-Margin Attack (MM attack) by using Wide Residual Network-34 (WRN34) as the backbone. MM attack is a strong adversarial attack that typically outperforms AutoAttack by focusing on maximizing the margin between the decision boundary and adversarial examples, making it particularly effective at exposing model vulnerabilities. We tested PGD-AT, PGD-AT with SD regularization (PGD-AT+SD), TRADES, and TRADES with SD regularization (TRADES+SD) on both CIFAR10 and CIFAR100 datasets. The results, including the average accuracy and standard deviation over 5 runs from different random seed, are shown in the table below:
>
> | **Method**     | **CIFAR10**       | **CIFAR100**      |
> | -------------- | ----------------- | ----------------- |
> | PGD-AT         | 50.24% ± 0.26     | 23.91% ± 0.15     |
> | PGD-AT+SD      | 52.29% ± 0.19     | 26.79% ± 0.22     |
> | TRADES         | 51.92% ± 0.32     | 21.34% ± 0.53     |
> | TRADES+SD      | 52.89% ± 0.28     | 23.45% ± 0.19     |
>
> Based on the table, the addition of SD still consistently improves the robust accuracy for both CIFAR10 and CIFAR100 datasets across different adversarial training methods. The integration of SD enhances the overall performance under Maximum-Margin Attack ($MM9$) conditions, indicating that SD effectively strengthens model robustness against adversarial attacks in various settings. These results suggest that the application of SD can serve as a beneficial regularizer, complementing standard adversarial training techniques to achieve better defensive outcomes.
>
> ## C2: More experiments on various attack budgets.
>
> To further evaluate the effectiveness of SD, we conducted experiments under PGD20 and CW with different attack budgets in Section 6.4 of our paper. These further results still show that our proposed SD can outperform the other counterparts.
>
> ## C3: Hyper-parameter concerns
>
> Please kindly allow us to clarify that **"hyperparameters in Table 1" is "NOT tricky"**.
>
> In Table 1 of our main paper, \(d\) and \(k\) are normalization parameters that are measured during training as shown in Algorithm 1 rather than selected; $\gamma$, $\omega$, and $\lambda$ work as internal parameters that are selected by grid search and not related with training methods of baselines. Regarding the training epochs and batch size, almost all comparison methods, such as RAT, LAS, and SCR, are different. Moreover, it is common for these to vary depending on the baseline and dataset ([1] [2] [3]). Other parameters, such as learning rate and weight decay, are kept consistent with the baselines for a fair comparison.
>
> In short, we follow the commonly used hyper-parameter selection strategy in machine learning that would typically ensure a fair comparison among different methods. We believe our produced improvements are consistent, independent of the baselines and do not rely on hyper-parameter tuning for baselines.
>
> [1] Jin, G., Yi, X., Wu, D., Mu, R., & Huang, X. (2023). Randomized adversarial training via taylor expansion. In Proceedings of the IEEE/CVF Conference on Computer Vision and Pattern Recognition (pp. 16447-16457).
>
> [2] Qian, Z., Zhang, S., Huang, K., Wang, Q., Yi, X., Gu, B., & Xiong, H. (2024). Perturbation diversity certificates robust generalization. Neural Networks, 172, 106117.
>
> [3] Zhang, S., Qian, Z., Huang, K., Wang, Q. F., Gu, B., Xiong, H., & Yi, X. (2024). Inter-feature Relationship Certifies Robust Generalization of Adversarial Training. International Journal of Computer Vision, 1-17.

---

### Review · Reviewer_GQUJ · 2024-10-08

**Summary Of Contributions:**

This paper studies the problem of robust generalization for representation learning.

- It proves a new generalization bound for the robust generalization problem.

- It links the quantities appearing in the bound to the volume of the latent spaces.

- Uses this insight to propose a practical algorithm, which is evaluated on a number of datasets.

**Audience:**

Yes

**Claims And Evidence:**

No

**Requested Changes:**

### Minors
- The one before last equation on page 23 is not correct. It should be a plus and not a minus between the two sums (you are implicitely using the triangular inequality when using the Lipschitz assumption).
- Please use the correct latex commands for scalar products in the proof of Thm. 3.2.
- In the proof of Thm. 3.2, detail why $tr(G) = N$.

### Major
- Add a discussion on the difference with Zhang et al. 2024.
- Adress the issues with both theorems as detailed above.
- Address the reproducibility issues.

**Strengths And Weaknesses:**

## Strengths
- Novel and interesting approach to the problem of robust generalization.
- Extensive evaluation and impressive results.

## Weaknesses
### Issues with Thm. 3.1
*Assumptions need to be clearly stated*
- The Lipschitz assumption needs to be explicited as it is done in the proof (and it should be used as an assumption in the proof, not kind of reintroduced)
- $L_d$ and $L_u$ need to be defined in the theorem (or say "They are constants $L_d$ and $L_u$ such that...")
- The definition of $M$ needs to be made clearer.

*Proof needs to be clearer*
- I do not understand the argument at the top of page 24 that is bowrrowed from Zhang et al. 2024 with the Lagrange mean value theorem, and it does not seem right to me.
- I do not understand why the Breteganolle-Huber-Carel inequality is needed. The term
$
\sum_{i = 1}^K (\frac{N_i}{N} - \mu(C_i))
$ is always zero, isn't it?
Same for
$
\sum_{i = 1}^K (\mu(C_i) - \frac{e}{N})
$.
In other words, this theorem is not a generalization bound, it is only relating the robust generalization to the generalization error via a decomposition of the error without any probabilistic argument.

*Very similar to Thm. 3.1 in Zhang et al. 2024*

There is a lack of comparison to the results of Zhang et al. 2024. From what I understand, the main difference with Zhang 2024 is that the bound here does not only concern the training error but also the generalization error. However, the resulting additional term $\nabla T_d$ is dismissed quite early in the paper.

### Issues with Thm. 3.2
- The statement of the theorem is not formal and as such not clear to me.
- Its status is not very clear: in the proof approximations are used, which is fine if they are discussed and if the final result is clearly marked as approximate.

### Issues with the algorithm
- Why is Gaussian noise added in the algorithm?
- The experimental section is very impressive. However, there are not many details: how were hyperparamters tuned? how does it compare to the tuning of the hyperparameters of the baselines? How many runs were performed? etc... Having access to the code would be helpful as well.

---

> ### Author Response · Authors · 2024-10-21
>
> ### Weakness in Theorem 3.1
>
> #### Lipschitz Assumption
>
> Thank you for your valuable feedback. We have now clearly stated the two Lipschitz assumptions before the proof, ensuring the logical flow is coherent and aligned with the formal structure.
>
> #### $L_d$ and $L_u$
>
> Thank you for pointing out the error on p.24, and the issues with $L_d$ and $L_u$.
>
> We have carefully considered your suggestion and have decided not to use the constants $L_d$ and $L_u$. The main reason is that these constants are difficult to express in a concrete analytical form, which might confuse the theoretical derivation. To avoid this issue, we revised the proof and introduced a new proof approach that simplifies and clarifies the derivation. Under the revised proof, we believe the above two weaknesses can be solved.
>
> Specifically, we made improvements to the original proof on p.25 and derived a new form through derivation in the revision of P. 26.
>
> This new derivation is clearer and avoids the use of constants that are difficult to express. We believe this improvement not only enhances the theoretical interpretability but also strengthens the rigor of the analysis.
>
> ### The definition of $M$
>
> In our study, we present $M$ in the same manner as prior works. Specifically, we define the loss upper bound $\underset{x^{adv} \in X^{adv}}{\sup} f(x^{adv})$ as $M$, where $X^{adv}$ represents the distribution of adversarial examples corresponding to the underlying data distribution. We have added this expression in the proof.
>
> ### Argument borrowed from Zhang et al. 2024
>
> We acknowledge the validity of your concern. The Mean Value Theorem may have certain limitations in our context. However, we still believe that $L_d$ and $L_u$ exist, similar to Lipschitz constants, although we are unable to provide an exact form. Therefore, we have modified the original derivation, as mentioned in the above response of $L_d$ and $L_u$.
>
> ### Breteganolle-Huber-Carel inequality
>
> Regarding your concern about whether $\left|\frac{N_i}{N} - \mu(C_i)\right|$ or $\left|\frac{e}{N} - \mu(C_i)\right|$ is always zero, we humbly think this is not the case. There might be a misunderstanding here because we're misusing $N_i$ and $|N_i|$.
>
> To solve this problem, we have replaced all $N_i$ with $|N_i|$ in the proof. In our context, $\mu(\mathcal{C}_i)$ denotes the probability of the $i$-th class' samples occurring in the underlying data distribution, while $(|N_1|, \cdots, |N_K|)$ is an i.i.d. multinomial random variable with parameters $N$.
>
> The term $\sum_{i=1}^{K} \left( \frac{|N_i|}{N} - \mu(C_i) \right)$ is not always zero because $\frac{|N_i|}{N}$ is a random variable that fluctuates with the sample distribution. Therefore, the Bretagnolle-Huber-Carel inequality is necessary to quantify this fluctuation and obtain a rigorous generalization bound.
>
> Such a format is widely used in existing RG studies [1,2]. The application of $\frac{e}{N}$ in the Bretagnolle-Huber-Carel inequality is well justified, despite it being a constant.
>
> While we believe our approach is valid and yields the desired form, we agree with your observation that the theorem could benefit from a more rigorous probabilistic foundation. In response, we have revised both the proof and the theorem, replacing $\frac{e}{N}$ with $\frac{|N_i|}{N}$ to incorporate the probabilistic argument you kindly suggested. We believe these changes enhance the clarity and correctness of the result.
>
>
>
> [1] Xu, H., & Mannor, S. (2012). Robustness and generalization. Machine learning, 86, 391-423.
>
> [2] Zhang, S., Qian, Z., Huang, K., Wang, Q., Zhang, R., & Yi, X. (2021, July). Towards better robust generalization with shift consistency regularization. In International conference on machine learning (pp. 12524-12534). PMLR.

---

> ### Author Response · Authors · 2024-10-21
>
> ## Weakness in Theorem 3.1(2)
> ### Compare with method of Zhang et al. 2024
>
> **Discard $||\nabla \boldsymbol{T}_d||_2$ early:**
>
> Regarding your instruction that we discard $||\nabla \boldsymbol{T}_d||_2$ early, it seems there has been a misunderstanding regarding our work.
> We assume you meant that we omit $||\nabla  \boldsymbol{T}||_2$ during the algorithm's implementation.
>
> In fact, we also did not ignore $||\nabla  \boldsymbol{T}||_2$ on the underlying data distribution, but assumed that this term will change
> with the $||\nabla \boldsymbol{T}_d||_2$ on the training data distribution, since the underlying data distribution is unavailable in the real world.
>
>
>
> In other words, $||\nabla \boldsymbol{T}_d||_2$ decreases with $||\nabla \boldsymbol{T}||_2$, i.e., $\underset{\theta}{\min} ||\nabla \boldsymbol{T}_d||_2 \rightarrow \underset{\theta}{\min} ||\nabla \boldsymbol{T}||_2$.
>
> This assumption is reasonable because it follows a basic assumption of machine learning methods — statistically, improving the feature representation on the available training set is essential and can often be assumed to generalize to improving the learned feature representation on the unknown underlying data distribution. It is noted that, in order to bridge the unknown underlying data with the available training data, prior works in adversarial robustness and representation learning have used similar assumptions, which lend support to our approach [1,2,3].
>
> Meanwhile, $||\nabla \boldsymbol{T}||_2$ is determined by both inter-class distance and intra-class distance. Our goal is to increase inter-class distance while simultaneously decreasing intra-class distance on the training set, which fosters more discriminative feature representations. Such adjustments on the training set are likely to reflect similar trends in the entire data distribution.
>
> **Compare with method of Zhang et al:**
>
> The approach proposed by Zhang et al. focuses on preserving the inter-feature relationship between natural and adversarial examples. This method aims to maintain the original feature distribution structure, minimizing the variation caused by adversarial perturbations.
>
> In contrast, our approach introduces the GMD to address the inconsistency between clean and adversarial examples in the inter-sample relationships. By optimizing a diverging spanned latent space, we enhance the discriminative power of feature representations.
>
> In the revision, we have added remarks and discussed the method of Zhang et al. in our revised paper.
>
>
> [1] Jin, G., Yi, X., Wu, D., Mu, R., & Huang, X. (2023). Randomized adversarial training via taylor expansion. In Proceedings of the IEEE/CVF Conference on Computer Vision and Pattern Recognition (pp. 16447-16457).
>
> [2] Qian, Z., Zhang, S., Huang, K., Wang, Q., Yi, X., Gu, B., & Xiong, H. (2024). Perturbation diversity certificates robust generalization. Neural Networks, 172, 106117.
>
> [3] Zhang, S., Qian, Z., Huang, K., Wang, Q. F., Gu, B., Xiong, H., & Yi, X. (2024). Inter-feature Relationship Certifies Robust Generalization of Adversarial Training. International Journal of Computer Vision, 1-17.

---

> ### Author Response · Authors · 2024-10-21
>
> ## Weakness in Thm. 3.2
>
> ### Not formal and clear
> Thanks for your valuable feedback. Per your kind requirements, we have thoroughly revised Theorem 3.2 and its proof. Please check the revised version.
>
> ## Weakness in algorithm
>
> ### Gaussian noise
> We add Gaussian noise to the initial images before generating adversarial examples mainly to increase data diversity and enhance the effectiveness of adversarial example generation. The Gaussian noise introduces randomness, expanding the coverage of training data and reducing overfitting. It is also a widely used protocol in previous studies [1,2,3].
>
> ### Experimental detail and code
>
> 1. **The number of parameters tuned:** We only use grid search to tune three hyperparameters: $\omega$, $\gamma$, and $\lambda$.
>
>    However, $\gamma$, $\omega$, and $\lambda$ work as internal parameters that are selected by grid search and are not related to training methods of baselines. Other parameters, such as learning rate and weight decay, are kept consistent with the baselines for a fair comparison.
>
> 2. **The number of runs:** Our results and standard deviations are obtained from 5 runs, each trained using a different random seed.
>
> 3. **Code of Implement:** Our code can be found in [SD code](https://anonymous.4open.science/r/FS-SD-FCB5/)
>
> ## Minors
>
> ### The last equation on page 23 and scalar products in the proof of Thm. 3.2
>
> Thanks for your feedback. We believe these issues have already been addressed after improving our proof.
>
> ### $tr(G) = N$
> Regarding your question on why $\text{tr}(\boldsymbol{G}) = N$, we express the forms of $\boldsymbol{D}$ and $\boldsymbol{D^*}$ to better illustrate this relationship. Specifically, we present the following expressions to provide a clearer understanding of the structure and behavior of these matrices in the context of our revision proof (P.27).
>
> ### Discussion on the difference with Zhang et al. 2024
>
> We have addressed this concern in the last response in Weakness in Thm. 3.1.
>
> ### Issues in theorems
>
> We have corrected all your concerns in theorems.
>
> ### Reproducibility issues
>
> We have uploaded our codes in  [SD code](https://anonymous.4open.science/r/FS-SD-FCB5/)
>
> [1] Wang, J., & Zhang, H. (2019). Bilateral adversarial training: Towards fast training of more robust models against adversarial attacks. In Proceedings of the IEEE/CVF international conference on computer vision (pp. 6629-6638).
>
> [2] Qian, Z., Zhang, S., Huang, K., Wang, Q., Yi, X., Gu, B., & Xiong, H. (2024). Perturbation diversity certificates robust generalization. Neural Networks, 172, 106117.
>
> [3] Zhang, H., & Wang, J. (2019). Defense against adversarial attacks using feature scattering-based adversarial training. Advances in neural information processing systems, 32.

---

> ### Author Response · Authors · 2024-10-21
> **More proof details**
>
> Dear Reviewer,
>
> Thank you for your insightful comments. Regarding your question about "why $tr(G) = N$" and our proof of the changes in the forms of $L_d$ and $L_u$, **due to Markdown rendering issues in OpenReview, we have included the detailed derivation and explanation in the supplementary material to ensure that the formulas and proofs are presented correctly.** We hope this supplementary content provides a clearer response to your concerns.
>
> Thank you again for your valuable feedback!
>
> Best regards,

---

### Author Response · Authors · 2024-10-21
**Response to Reviewer Feedback and Revisions Summary**

Dear Reviewers,

Thank you for your professional and valuable feedback. We have addressed all the questions in the rebuttal.

For the theoretical concerns raised by Reviewer aW6S and Reviewer GQUJ, we have made the necessary revisions. Additionally, **due to markdown limitations in OpenReview, we have included the detailed proofs in the supplementary materials for better presentation.**

We have also added the experimental supplements as suggested by Reviewer thck in the revised manuscript.

**Please Note that all changes in the revision have been highlighted in blue for easy reference.**

We look forward to receiving further feedback and suggestions to improve our work. Thank you again for your insightful comments.

Best regards,

---

### Decision · Action_Editor_nSNh · 2024-11-20

**Recommendation:** Accept as is

**Comment:**

The reviewers recognized the paper's rigorous methodology and robust experiments against both mainstream and stronger adversarial attacks. They appreciated the thorough rebuttal that addressed initial concerns by clarifying theoretical aspects and expanding analyses. One minor concern raised by reviewers is the term "certifying" in the title, which they found disconnected from the content.

**Audience:**

TMLR audience would be interested in knowing the findings of this paper. The paper explores the robust features of adversarial training methods, a topic that holds significant relevance in the field of machine learning. The paper offers valuable insights that could benefit researchers and practitioners focused on improving the robustness of machine learning models against adversarial attacks.

**Claims And Evidence:**

The methodology is described as rigorous, with observational experiments verifying the realistic feasibility of the analysis. Additionally, the evaluation addresses a comprehensive range of attacks, including mainstream ones such as PGD and AA, and demonstrates robustness even under stronger attacks. These experiments sufficiently validate the advancement of the proposed method over other adversarial training improvements.